# Bernstein–von Mises for Adaptively Collected Data

**Kevin Du**
Department of Statistics
Harvard University
kevindu@college.harvard.edu

**Yash Nair**
Department of Statistics
Stanford University
yashnair@stanford.edu

**Lucas Janson**
Department of Statistics
Harvard University
ljanson@fas.harvard.edu

## Abstract

Uncertainty quantification (UQ) for adaptively collected data, such as that coming from adaptive experiments, bandits, or reinforcement learning, is necessary for critical elements of data collection such as ensuring safety and conducting after-study inference. The data's adaptivity creates significant challenges for frequentist UQ, yet Bayesian UQ remains the same as if the data were independent and identically distributed (i.i.d.), making it an appealing and commonly used approach. Bayesian UQ requires the (correct) specification of a prior distribution while frequentist UQ does not, but for i.i.d. data the celebrated Bernstein–von Mises theorem shows that as the sample size grows, the prior 'washes out' and Bayesian UQ becomes *frequentist*-valid, implying that the choice of prior need not be a major impediment to Bayesian UQ as it makes no difference asymptotically. This paper for the first time extends the Bernstein–von Mises theorem to adaptively collected data, proving asymptotic equivalence between Bayesian UQ and Wald-type frequentist UQ in this challenging setting. Our result showing this asymptotic agreement does not require the standard stability condition required by works studying validity of Wald-type frequentist UQ; in cases where stability is satisfied, our results combined with these prior studies of frequentist UQ imply frequentist validity of Bayesian UQ. Counterintuitively however, they also provide a negative result that Bayesian UQ is *not* asymptotically frequentist valid when stability fails, despite the fact that the prior washes out and Bayesian UQ asymptotically matches standard Wald-type frequentist UQ. We empirically validate our theory (positive and negative) via a range of simulations.

## 1 Introduction

Data in applications such as robotics [19], healthcare [32], clinical trials [5], online education [27], mobile health [2, 31], and online advertising [22, 10] is routinely being collected *adaptively*. This means that the data is collected sequentially, with decisions about the data collection itself being made online based on all the data observed up to that time point. In particular, the online decisions often try to focus on actions or interventions that prior data indicates will produce large values of some reward function, and this type of data collection includes adaptive experiments, multi-armed bandits, and reinforcement learning.

Two important elements of adaptive data collection are (1) assessing at the end of the data collection what has been learned (e.g., to assess confidence in a putatively optimal clinical treatment or inform a future online advertising campaign) [33, 14] and (2) making sure to avoid certain bad outcomes

39th Conference on Neural Information Processing Systems (NeurIPS 2025).

during the data collection process (e.g., avoiding crashing a robot while it is reinforcement learning) [4]. Both of these elements, as well as others, rely critically on *uncertainty quantification* (UQ) of the parameters of the data-generating environment, yet standard frequentist UQ, such as Wald-type inference based on the maximum likelihood estimator (MLE), is made far more complicated by the adaptivity of the data collection [21] which results in data which is not i.i.d., Markovian, or even stationary.

On the other hand, the Bayesian statistical paradigm translates seamlessly to the adaptive setting, so that the vast wealth of Bayesian UQ methodology designed for i.i.d. data can be directly applied to adaptively collected data to provide strong Bayesian probabilistic guarantees. However, the validity of those guarantees is predicated on correctly specifying a prior distribution for the parameters, which can be challenging to do or justify in practice. For i.i.d. data, the celebrated Bernstein–von Mises (BvM) theorem [29, Theorem 10.1] proves that for any reasonable choice of prior, Bayesian UQ asymptotically matches that of prior-free Wald-type frequentist UQ, providing strong Bayesian and frequentist justification for Bayesian UQ in large samples *without* the analyst having to worry too much about the choice of prior. Although BvM has been extended beyond i.i.d. data in a number of ways (see Section 2 for more details), it has not been extended to the setting of adaptively collected data.

**Contributions** This paper for the first time proves BvM results for adaptively collected data, showing that Bayesian UQ asymptotically matches (asymptotically normal) Wald-type frequentist UQ under certain mild conditions. Our first result, in Section 3, applies to a very general class of adaptive linear Gaussian settings that include Gaussian multi-armed bandits, adaptive Gaussian linear bandits (a form of contextual bandit), and the linear quadratic regulator (a form of reinforcement learning on a Markov decision process). We then show in Section 4 that in the special case of multi-armed bandits, the conditions of our first result can be weakened, and the Gaussian assumption can also be generalized to any exponential family. Finally, we show in Section 5 that the conditions of the first result can also be weakened for linear (contextual) bandits. Section 6 empirically validates our theoretical results. A surprising aspect of our BvM results is that they do not require the key stability condition used in frequentist validity results; however, it is known that Wald-type frequentist inference may fail to be asymptotically valid in cases when the stability condition does not hold. The counterintuitive implication is that in some adaptive settings, Bayesian UQ asymptotically matches frequentist UQ (and the prior becomes irrelevant), yet Bayesian UQ is *not* asymptotically frequentist-valid.

**Notation and Terminology.** Throughout this paper, we refer to a confidence set as *asymptotically frequentist-valid* at level $\alpha$ if for any parameter value, the limit inferior of the frequentist coverage is at least $1 - \alpha$. Similarly, we refer to a credible set as *asymptotically Bayesian-valid* at level $\alpha$ if the limit inferior of the Bayesian coverage is at least $1 - \alpha$. We will use the term frequentist (resp. Bayesian) UQ generically to refer to any of the many forms of frequentist (resp. Bayesian) statistical inference such as hypothesis tests, confidence intervals, or prediction intervals (resp. such as Bayesian hypothesis tests, credible intervals, or posterior predictive intervals). By *Wald-type frequentist UQ*, we refer to standard asymptotic hypothesis testing or confidence interval construction using the MLE and its asymptotic Normality—for instance, as demonstrated in Example 15.6 of [29]. Let $\lambda_{\min}(A)$ and $\lambda_{\max}(A)$ denote the minimum and maximum eigenvalues of the matrix $A$ respectively.

## 2 Background

As mentioned in the previous section, Bayesian statistical inference requires no adjustment for the adaptivity of the data. This both makes the Bayesian approach an appealing and commonly used tool for UQ in adaptively collected data and means there is no need for (and thus a lack of) prior work extending it to the adaptive setting. Thus, prior work on UQ for adaptively collected data has focused on frequentist UQ, with the primary approach being to make assumptions on both the data and the adaptive assignment algorithm by which it is collected. These assumptions either (1) enable the use of martingale central limit theorems to establish asymptotic normality of common frequentist estimators like the MLE (see, e.g., [21, 33, 14, 6, 12, 23]) or (2) allow for conservative finite-sample inference via martingale-based concentration bounds [15, 16, 1]. These conditions are often highly technical and hard to check, and reduce to or are related to a *stability* condition introduced in [21]; see Section 3 for more details. Another line of work uses randomization testing for frequentist UQ

for adaptively collected data, providing non-asymptotic and non-conservative guarantees but relying on sampling procedures that can be computationally prohibitive to use [25].

The classical BvM theorem [29, Theorem 10.1] proves that under minimal conditions, for i.i.d. data drawn from a parametric model, the posterior distribution is asymptotically normal centered at the MLE with variance equal to the inverse Fisher information, and hence agrees asymptotically with (and inherits the asymptotic frequentist validity of) Wald-style frequentist inference. Prior works generalizing BvM to non-i.i.d. data only apply in settings where the ratio of the maximum and minimum eigenvalues of the Fisher information matrix is bounded [18, 11, 8, 20, 7, 9]. This condition fails to hold in most adaptive settings where, over time, certain actions are learned to be better than others and are asymptotically sampled infinitely more often than the worse actions, resulting in an unbounded ratio (e.g., regret-optimal multi-armed bandit algorithms sample suboptimal arms logarithmically often, leading to an eigenvalue ratio that grows like $n/\log(n)$ [3]).

## 3    BvM on Adaptive Linear Gaussian Data

This paper will primarily consider the adaptive data collection setting, laid out in Algorithm 1, that at time step $j$ allows the data collector to choose via an arbitrary function $\Lambda$ a covariate vector $x_j$ based on all data observed so far ($H_{j-1} = ((x_1, y_1), \ldots, (x_{j-1}, y_{j-1}))$), and then observe $y_j \sim \mathcal{N}(x_j^\top \beta, \sigma^2)$. We assume the only unknown in this sampling procedure is $\beta$, and hence the task at hand is to perform UQ for $\beta$ based on the full data trajectory $H_n = (\mathbf{X}_n, \mathbf{y}_n)$, where $\mathbf{X}_n$ is the $n \times p$ matrix with $j$th row given by $x_j^\top$ and $\mathbf{y}_n$ is the $n$-vector with $j$th entry given by $y_j$.

---

**Algorithm 1** Adaptive Linear Gaussian Sampling Procedure

---

**Input** Covariate sampling rule $\Lambda : (\mathbb{R}^p \times \mathbb{R})^* \to \Delta(\mathbb{R}^p)$, coefficient vector $\beta \in \mathbb{R}^p$, variance $\sigma^2 \in \mathbb{R}^+$
**Output** Sampling trajectory $H_n$

$H_0 \leftarrow \emptyset$
**for** $j = 1, \ldots, n$ **do**
    Sample $x_j \sim \Lambda(\cdot | H_{j-1})$
    Sample $y_j \sim \mathcal{N}(x_j^\top \beta, \sigma^2)$
    $H_j \leftarrow ((x_1, y_1), \ldots, (x_j, y_j))$
**end for**

---

Here, $\Lambda$ denotes a placeholder that can represent any sampling algorithm including UCB, Thompson sampling, autoregressive models, etc. We now present our first result, which states that the posterior distribution is asymptotically normal, centered at the MLE with variance equal to the inverse empirical Fisher information.

**Theorem 1.** *Suppose $\pi(\beta)$ is the prior distribution for $\beta$ and $\pi(\beta|H_n)$ is the posterior after observing the trajectory $H_n$. Let $\hat{\beta}_n = (\mathbf{X}_n^\top \mathbf{X}_n)^{-1} \mathbf{X}_n^\top \mathbf{y}_n$ be the MLE for $\beta$ and let $\beta_0$ be the true value of $\beta$. Assume the sampling procedure in Algorithm 1 satisfies the following conditions:*

*(i)* $\lambda_{\min}(\mathbf{X}_n^\top \mathbf{X}_n) \xrightarrow{p} \infty$ *and* $\log \lambda_{\max}(\mathbf{X}_n^\top \mathbf{X}_n) = o_p(\lambda_{\min}(\mathbf{X}_n^\top \mathbf{X}_n))$.

*(ii)* $\pi(\cdot)$ *is continuous with positive density at* $\beta_0$.

*Then, the posterior distribution $\pi(\beta|H_n)$ satisfies*

$$\|\pi(\beta|H_n) - \mathcal{N}(\hat{\beta}_n, \sigma^2(\mathbf{X}_n^\top \mathbf{X}_n)^{-1})\|_{\mathrm{TV}} \xrightarrow{p} 0.$$

Theorem 1's proof, given in Appendix B.1, first upper bounds the key total variation (TV) distance in terms of the $L^1$ distances of *unnormalized* densities, allowing us to ignore normalizing constants in our analysis. Our proof then uses the fact that the posterior distribution is not affected by the adaptivity of the algorithm, since terms in the likelihood involving the covariate sampling process $\Lambda$ do not depend on the parameter and can thus be absorbed into the normalizing constant. We also use a common trick in BvM-style proofs [18, 29] of truncating the posterior to local ellipsoids defined as

$\{\beta : \|(\mathbf{X}_n^\top \mathbf{X}_n)^{1/2}(\beta - \beta_0)\| \le M_n\}$ for some local radius $M_n$. This truncation, however, differs from those used in previous BvM-style proofs, as here the empirical Fisher information $\mathbf{X}_n^\top \mathbf{X}_n / \sigma^2$ does not necessarily have bounded condition number, implying that the resulting ellipsoids may be highly anisotropic, stretching much wider in some directions than in others. This step of our proof relies critically on the second part of condition *(i)* of Theorem 1 which allows us to show that the density of the representative normal distribution outside of local ellipsoids converges to zero. We furthermore show that the posterior can be written as proportional to the prior multiplied by the density of the representative normal, meaning we can truncate the distribution to a local ellipsoid even if the prior is unbounded.

Our proof requires condition *(i)* for a couple of reasons. If the maximum eigenvalue of the observation matrix grows exponentially faster than the minimum eigenvalue, there are two potential pathological consequences. First, Lai and Wei showed that condition *(i)* of Theorem 1 is nearly necessary for MLE consistency; in particular, if condition *(i)* of Theorem 1 is removed, the MLE is not necessarily consistent [21, Example 1]. Second, even if $\hat\beta_n$ is consistent but condition *(i)* fails, it may no longer be the case that the density of the distribution $\mathcal{N}(\hat\beta_n, \sigma^2(\mathbf{X}_n^\top \mathbf{X}_n)^{-1})$ at some other point $\beta' \ne \beta_0$ converges to $0$. One can see this by writing the density as

$$\mathcal{N}(\beta'; \hat\beta_n, \sigma^2(\mathbf{X}_n^\top \mathbf{X}_n)^{-1}) = \sqrt{\frac{|\mathbf{X}_n^\top \mathbf{X}_n|}{(2\pi\sigma^2)^p}} \exp(-\frac{1}{2\sigma^2}(\beta' - \hat\beta_n)^\top \mathbf{X}_n^\top \mathbf{X}_n(\beta' - \hat\beta_n)),$$

where $\mathcal{N}(x; \mu, \sigma^2)$ denotes the density value at $x$ of the $\mathcal{N}(\mu, \sigma^2)$ distribution. While the exponential term in the above expression necessarily converges to zero if $\beta' \ne \beta_0$, $\hat\beta_n$ is consistent, and $\lambda_{\min}(\mathbf{X}_n^\top \mathbf{X}_n) \xrightarrow{p} \infty$, it is not necessarily true that the entire expression converges to zero or is even bounded. This is because the quadratic term inside the exponential may only be of order $O(\lambda_{\min}(\mathbf{X}_n^\top \mathbf{X}_n))$ whereas the term $|\mathbf{X}_n^\top \mathbf{X}_n|$ grows at least as quickly as $\lambda_{\max}(\mathbf{X}_n^\top \mathbf{X}_n)$. Thus, although the measure of a neighborhood of $\beta'$ under the probability measure $\mathcal{N}(\hat\beta_n, \sigma^2(\mathbf{X}_n^\top \mathbf{X}_n)^{-1})$ must converge to zero, without condition *(i)*, it is possible that the *density* of this distribution diverges at $\beta'$. We show in the proof of Theorem 1 that the likelihood function is proportional to the density of $\mathcal{N}(\hat\beta_n, \sigma^2(\mathbf{X}_n^\top \mathbf{X}_n)^{-1})$, meaning that it might be possible that the density of both the prior and the normalized likelihood function diverge at $\beta'$. This could potentially cause the posterior to have asymptotically non-negligible measure in a neighborhood of $\beta'$, which would violate the BvM statement as the measure of the normal distribution $\mathcal{N}(\hat\beta_n, \sigma^2(\mathbf{X}_n^\top \mathbf{X}_n)^{-1})$ converges to zero in a neighborhood of $\beta'$.

Notably, Theorem 1 does not require the key stability condition, originally from [21] but used in many works since then for frequentist UQ for adaptively collected data, which assumes the existence of a deterministic sequence $\mathbf{B}_n$ for which $\mathbf{B}_n^{-1}(\mathbf{X}_n^\top \mathbf{X}_n)^{1/2} \xrightarrow{p} I_{p \times p}$. Without this condition, $\hat\beta_n$ may fail to be asymptotically normal [21, Example 3]. In Proposition 2 in Appendix D we show that their example *does*, however, satisfy the conditions of our Theorem 1. This leads to the rather surprising result that there are settings where Bayesian UQ is asymptotically equivalent to standard Wald-style frequentist UQ, yet the latter (and therefore also the former) is asymptotically frequentist-invalid. We will describe another such example in the next section, which requires the triangular array version of BvM we prove there.

The previous paragraph notes that Theorem 1 can indicate either asymptotic frequentist validity or invalidity of Bayesian UQ, depending on the situation. However, in terms of Bayesian validity, Theorem 1 (and indeed all the theorems in the paper) provide a strong positive result regarding misspecified priors: they say that the prior distribution $\pi$ is 'washed out' as $n \to \infty$. Thus, from a Bayesian perspective, a credible interval asymptotically has the correct coverage even if the prior is misspecified, as long as both the correct prior and the misspecified prior used for the inference are continuous and bounded, with the misspecified prior's support containing that of the correct prior. Note, however, that the rate at which the "wash out" effect occurs depends on the rate at which the posterior distribution converges in TV distance to a normal distribution, which may be logarithmically slow for optimal bandit algorithms. Thus, it may not be accurate in finite samples to treat a misspecified prior as having "washed out".

## 4 BvM on Multi-armed Bandits

A notable special case of Algorithm 1 is the multi-armed bandit setting, which corresponds to the case when $\Lambda$'s output is supported only on the basis vectors. The multi-armed bandit setting can model experiments involving adaptively chosen treatment arms—typically by some optimization algorithm like UCB [3] or Thompson sampling [17]—where we want to estimate the mean outcome of each arm. For the bandit setting, we will use the notation $N_{i,n}, \hat{\mu}_i^{(n)}$ to denote the count and sample mean respectively of the pulls of arm $i$ in the first $n$ steps.

As discussed in the prior section, our BvM proof requires the assumption that the maximum eigenvalue of the data matrix grows subexponentially with respect to the minimum eigenvalue. However, this assumption is violated by many popular bandit algorithms such as UCB, where suboptimal arms are pulled only $O_p(\log n)$ times. Thus, we require a modification of the above proof to specifically the case of bandits to include these existing algorithms. We also show that we can generalize our result to triangular arrays of data—that is, we now allow the adaptive decision rule to depend on the sequence length $n$ and superscript it by $n$ to make this dependence explicit: $\Lambda^n$. However, we assume that the parameter $\beta_0$, variance $\sigma^2$, and prior $\pi$ remain the same throughout the array. The triangular array formulation enables us to extend our results to sampling procedures which have policies depending on the length of the overall experiment such as is the case in the batched bandit setting [33]. First, we show a consistency result for the case of triangular array bandits.

**Lemma 1.** *Suppose independent length-$m_n$ trajectories $H_{m_n}^n = ((x_1^n, y_1^n), \ldots, (x_{m_n}^n, y_{m_n}^n))$ are drawn via Algorithm 1 using sampling rules $\Lambda^n$ for each $n$, where $\beta$ and $\sigma^2$ remain the same for all trajectories. Let $\mathbf{X}_n = \begin{pmatrix} x_1^n & \cdots & x_{m_n}^n \end{pmatrix}^\top$ and $\mathbf{y}_n = \begin{pmatrix} y_1^n & \cdots & y_{m_n}^n \end{pmatrix}^\top$. Assume the triangular array version of Algorithm 1 satisfies the following conditions:*

*(i) Each $x_j^n$ is a basis vector, i.e. $x_j^n \in \{e_1, \ldots, e_p\}$.*

*(ii) $\lambda_{\min}(\mathbf{X}_n^\top \mathbf{X}_n) \xrightarrow{p} \infty$.*

*Then, the MLE $\hat{\beta}_n = (\mathbf{X}_n^\top \mathbf{X}_n)^{-1} \mathbf{X}_n^\top \mathbf{y}_n$ is consistent, i.e. $\hat{\beta}_n \xrightarrow{p} \beta_0$.*

The proof of Lemma 1 appears in Appendix C. In addition to this consistency result, we also need a condition that allows us to truncate the posterior distribution. Note that the theorem below assumes that the prior density $\pi$ is bounded which we did not need in the proof of Theorem 1; this condition guarantees that the posterior can be asymptotically approximated as proportional to the likelihood when the regularity condition on the maximum eigenvalue is not met. With these changes, we no longer need the condition that $\log \lambda_{\max}(\mathbf{X}_n^\top \mathbf{X}_n) = o_p(\lambda_{\min}(\mathbf{X}_n^\top \mathbf{X}_n))$, making Theorem 2 very broadly applicable to standard bandit algorithms.

**Theorem 2.** *Assume the triangular array version of Algorithm 1 satisfies the following conditions:*

*(i) Each $x_j^n$ is a basis vector, i.e. $x_j^n \in \{e_1, \ldots, e_p\}$.*

*(ii) $\lambda_{\min}(\mathbf{X}_n^\top \mathbf{X}_n) \xrightarrow{p} \infty$.*

*(iii) $\pi(\cdot)$ has continuous and bounded density on $\mathbb{R}^p$ which is positive at $\beta_0$.*

*Then, the posterior distribution $\pi(\beta | H_{m_n}^n)$ satisfies*

$$\|\pi(\beta | H_{m_n}^n) - \mathcal{N}(\hat{\beta}_n, \sigma^2 (\mathbf{X}_n^\top \mathbf{X}_n)^{-1})\|_{\text{TV}} \xrightarrow{p} 0.$$

**Remark 1.** *This result assumes homoskedasticity of the bandits, but it also holds if each arm has a different (but still known) variance $\sigma_1^2, \ldots, \sigma_p^2 > 0$, as shown in Theorem 5 in the Appendix E.*

The proof of Theorem 2 appears in Appendix B.2. Note that the rate of convergence of the TV distance in Theorem 2 depends on the growth rate of $\lambda_{\min}(\mathbf{X}_n^\top \mathbf{X}_n)$, which can be logarithmically slow for optimal bandit algorithms as seen in Section 6. Theorem 2 gives the following corollary (proven in Appendix D) in the setting of non-triangular-array bandits, showing that the BvM convergence statement is uniform over sampling rules as long as $\lambda_{\min}(\mathbf{X}_n^\top \mathbf{X}_n)$ grows arbitrarily large among these sampling rules.

**Corollary 1.** *For any sequences $(r_n, \epsilon_n)$ for which $r_n \to \infty$ and $\epsilon_n \to 0$, let $\mathcal{P}_n$ be the sequence of sets of distributions $P$ of trajectories induced by sampling rules $\Lambda$ such that for all $n$ and for all $P \in \mathcal{P}_n$,*

*(i) Each $x_j^n$ is a basis vector, i.e. $x_j^n \in \{e_1, \ldots, e_p\}$.*

*(ii) $P(\lambda_{\min}(\mathbf{X}_n^\top \mathbf{X}_n) > r_n) > 1 - \epsilon_n$.*

*Then, we have for any $c > 0$,*

$$\limsup_{n \to \infty} \sup_{P \in \mathcal{P}_n} P(\|\pi(\beta|H_n) - \mathcal{N}(\hat{\beta}_n, \sigma^2(\mathbf{X}_n^\top \mathbf{X}_n)^{-1})\|_{\text{TV}} > c) = 0.$$

Theorem 2 implies that the interval $[\hat{\mu}_i^{(n)} \pm z_{1-\alpha/2}\sigma N_{i,n}^{-1/2}]$ is an asymptotically valid Bayesian credible interval. Note that this matches the Wald-type frequentist interval which one uses in i.i.d. settings where the MLE is indeed normal. But [33] shows that for Thompson sampling in the two-arm batched bandit setting, the distribution of the sample means is not asymptotically normal in the case when the true arm means are equal. Thus, Theorem 2 implies that in this setting the credible interval will fail to be asymptotically frequentist-valid despite asymptotically matching the usual Wald-type frequentist confidence interval.

We can generalize this result beyond Gaussian bandits to exponential family bandits, i.e., where we still constrain $x_j$ in Algorithm 1 to be a basis vector, but now instead of $y_j$ being sampled from a Gaussian with mean $x_j^\top \beta$, it is sampled from a exponential family (with density $\exp(\eta y_j - b(\eta))h(y_j)$) with parameter $\eta = x_j^\top \beta$. Exponential family models include many common distribution types such as Bernoulli, Poisson, Gamma, Beta, and Geometric distributions. For the remainder of our results in this paper, we return to the non-triangular array setting where $\Lambda$ is not allowed to depend on $n$.

**Theorem 3.** *Let $\beta_0 \in \mathbb{R}^p$ be the true parameter value and $\beta_{0,i}$ be the $i$-th coordinate of $\beta_0$. Let $N_{n,i}$ be the number of times arm $i$ was pulled and $\bar{Y}_{n,i} = \frac{1}{N_{n,i}}\sum_{j=1}^n I[x_j = i]y_j$. Let the local MLE be $\hat{\beta}_{n,i} = \beta_{0,i} + \frac{\bar{Y}_{n,i} - b'(\beta_{0,i})}{b''(\beta_{0,i})}$ and the empirical Fisher information be $I_n = \text{diag}\{N_{i,n}b''(\beta_{0,i})\}$. Suppose the exponential family version of Algorithm 1 satisfies the following properties:*

*(i) $\beta_{0,1}, \ldots, \beta_{0,p}$ are in the interior of the natural parameter space.*

*(ii) $\min_i N_{n,i} \overset{p}{\to} \infty$.*

*(iii) $\pi(\cdot)$ has continuous and bounded density on $\mathbb{R}^p$ which is positive at $\beta_0$.*

*Then*

$$\|\pi(\beta|H_n) - \mathcal{N}(\hat{\beta}_n, I_n^{-1})\|_{\text{TV}} \overset{p}{\to} 0.$$

The proof of Theorem 3 appears in Appendix B.3. The local MLE defined above is the maximizer of the second order Taylor expansion of the log-likelihood around $\beta_0$, and it is asymptotically equivalent to the true MLE by the asymptotic equivalence of the log-likelihood to its second-order Taylor expansion in a local neighborhood of the true parameter value. Note that the above theorem requires the true parameters to be in the interior of the natural parameter space, meaning for extremes of common models such as Poisson with rate near 0 or Bernoulli with probability near 1, the conclusion of the theorem may be a bad approximation for finite $n$.

The proof of this theorem is similar to that of Theorem 1, but here we require a Taylor expansion to express the likelihood as a function proportional to a Gaussian density. The bandit setting is particularly nice for performing this expansion since the likelihood factors into the product of the likelihoods for each individual arm. Thus, even if the eigenvalues of $I_n$ grow at asymptotically different rates, the likelihood is still well-approximated by the second-order expansion. This argument is harder when generalizing beyond bandits, as argued in Appendix F. Additionally, to argue that we can truncate the posterior distribution to a local neighborhood, we use the convexity of $b(\cdot)$ to bound the tails of the likelihood function. Our proof does not necessarily generalize to triangular arrays as it relies on uniform bounds on a serialized sequence of samples, such as the one provided by the law of iterated logarithms.

## 5    BvM on Gaussian Linear Bandits

Another special case for our BvM theorem is the case of Gaussian linear bandits, in which a context $x_j$ is observed for each arm pull $a_j$, impacting the resulting reward distribution through a linear transformation, i.e. $y_j \sim \mathcal{N}(x_j^\top \theta_{a_j}, \sigma^2)$ where $\theta_1, \ldots, \theta_m$ are parameter vectors. To model contextual bandits with our adaptive linear Gaussian data process in Algorithm 1, we will let the parameter be $\beta \in \mathbb{R}^{md}$ where the $(id - d + 1)$–$(id)$th indices of $\beta$ represent $\theta_i$; in other words, we stack the parameter vectors $\theta_1, \ldots, \theta_m$ vertically. Then, when we observe context $x' \in \mathbb{R}^d$ and action $i \in \{1, \ldots, m\}$, we sample the outcome from $\mathcal{N}(\beta^\top x, \sigma^2)$ where the $(id - d + 1)$–$(id)$th index of $x \in \mathbb{R}^{md}$ represents $x'$. Then, we can state BvM in the contextual bandit setting as follows.

**Theorem 4.** *Suppose $p = md$ and we decompose the covariate space as*

$$\mathcal{X} := \left(\mathbb{R}^d \times \{\mathbf{0}_d\} \times \cdots \times \{\mathbf{0}_d\}\right) \cup \left(\{\mathbf{0}_d\} \times \mathbb{R}^d \times \cdots \times \{\mathbf{0}_d\}\right) \cup \cdots \cup \left(\{\mathbf{0}_d\} \times \{\mathbf{0}_d\} \times \cdots \times \mathbb{R}^d\right)$$

*Assume the sampling procedure in Algorithm 1 satisfies the following conditions:*

*(i) Each action $x_j$ falls in $\mathcal{X}$.*

*(ii) For all $i = 1, \ldots, m$, let $T_i : \mathcal{X} \to \mathbb{R}^d$ be a projection onto the $(id - d + 1)$–$(id)$th coordinates. Letting $I_{n,i} = \sum_{j=1}^n T_i(x_j)T_i(x_j)^\top \in \mathbb{R}^{d \times d}$, for all $i$ we have*

$$\lambda_{\min}(I_{n,i}) \xrightarrow{p} \infty \text{ and } \log \lambda_{\max}(I_{n,i}) = o_p(\lambda_{\min}(I_{n,i})).$$

*(iii) $\pi(\cdot)$ is continuous with bounded density on $\mathbb{R}^p$ and positive density at $\beta_0$.*

*Then, the posterior distribution $\pi(\beta|H_n)$ satisfies*

$$\|\pi(\beta|H_n) - \mathcal{N}(\hat{\beta}_n, \sigma^2(\mathbf{X}_n^\top \mathbf{X}_n)^{-1})\|_{\mathrm{TV}} \xrightarrow{p} 0.$$

The proof of Theorem 4 appears in Appendix B.4. Note that condition (ii) in Theorem 4 is different from condition *(i)* in Theorem 1 because it concerns the distribution of contexts conditioned on a particular arm, rather than the distribution of arm pulls. Thus, it is possible for a contextual bandit algorithm to satisfy condition (ii) when it pulls different arms at exponentially different rates. This result does not immediately generalize to triangular arrays, since we rely on consistency which may not hold in the triangular array setting for contextual bandits. However, the non-triangular array version of Theorem 2 is a special case of Theorem 4.

## 6    Numerical Experiments

We discuss the empirical validity of the statement of Theorem 1 in the setting of multi-armed bandits, contextual bandits, and the linear quadratic regulator (LQR). As seen in the previous section, the posterior distribution is asymptotically equivalent in TV distance to the normal distribution $\mathcal{N}(\hat{\beta}_n, \sigma^2(\mathbf{X}_n^\top \mathbf{X}_n)^{-1})$, which we call the representative normal distribution. Although this equivalence holds asymptotically, the convergence rate to this representative normal distribution may be quite slow depending on the rate of growth of $\lambda_{\min}(\mathbf{X}_n^\top \mathbf{X}_n)$. We empirically perform posterior inference in three common adaptive settings and show that the posterior does empirically converge to the representative normal.

We use Monte Carlo and the relation $\|P - Q\|_{\mathrm{TV}} = \mathbb{E}_{X \sim P}\left[\max\left(0, 1 - \frac{Q(X)}{P(X)}\right)\right]$ to approximate the TV distance between the posterior distribution and the representative normal distribution. As seen in Figures 1, 2, and 3, the convergence rate of the TV distance depends on the configuration of the true parameters. Note that in the multi-armed bandit setting, the convergence seems fastest when the arm means are equal and becomes slower as the margin increases. One explanation for this is that the result of Theorem 2 requires the condition $\lambda_{\min}(\mathbf{X}_n^\top \mathbf{X}_n) \xrightarrow{p} \infty$, suggesting that the rate at which this minimum eigenvalue grows determines the rate of convergence of the TV distance. More specifically, we see that in our proof of Theorem 2, we use the expression $\mathbb{E}\left[\left(\frac{\pi(\kappa_n)}{\pi(\beta_0)} - 1\right)_+\right]$ as an

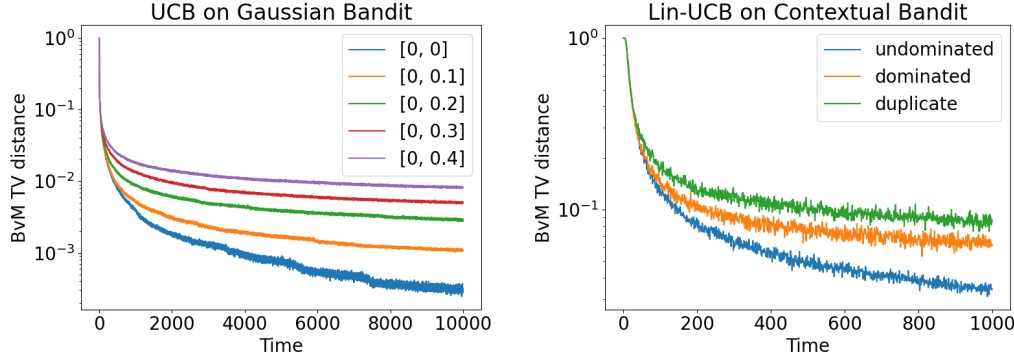

Figure 1: (Left) Average TV distance measured in the BvM statement for UCB in two-arm Gaussian bandits over horizon $T = 10^4$ using $10^4$ replicates under five different true parameter configurations labelled by $[\mu_1, \mu_2]$ where $\mu_1, \mu_2$ are the true means. (Right) Average TV distance measured in the BvM statement for lin-UCB on three-arm Gaussian linear contextual bandits with context distribution $\mathcal{N}(0, I_{2 \times 2})$ under three different true parameter configurations. Standard Gaussian priors are used for all arms. TV estimates shown have standard error at most $0.1$ times the TV estimate.

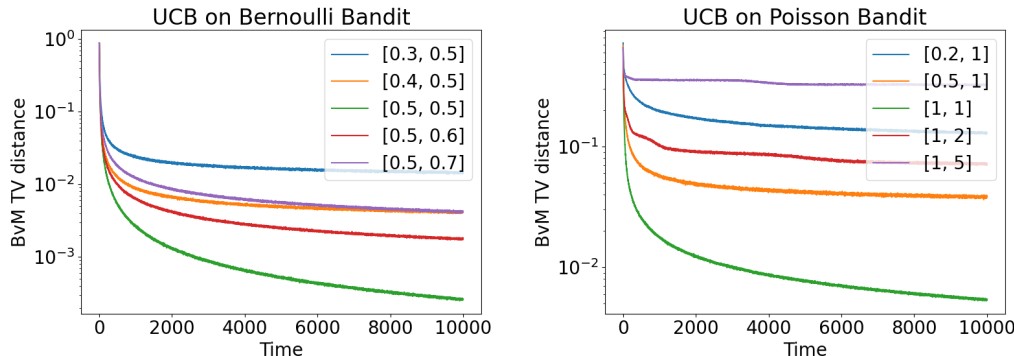

Figure 2: Average BvM TV distance for UCB on Bernoulli bandits and Poisson bandits, under the same configurations as Figure 1. Beta$(1, 1)$ priors are used for the Bernoulli bandit and Gamma$(1, 1)$ priors for the Poisson bandit. The representative normal is centered at the true MLE, which is asymptotically equivalent to the local MLE used in Theorem 3. TV estimates shown have standard error at most $0.1$ times the TV estimate.

upper bound on the TV distance where $\kappa_n | H_{m_n}^n \sim \mathcal{N}(\hat{\beta}_n, \sigma^2(\mathbf{X}_n^\top \mathbf{X}_n)^{-1})$. Thus, we see that the rate at which $\lambda_{\min}(\mathbf{X}_n^\top \mathbf{X}_n)$ diverges affects the convergence rate of $\kappa_n \xrightarrow{p} \beta_0$, which in turn affects the convergence $\frac{\pi(\kappa_n)}{\pi(\beta_0)} - 1 \xrightarrow{p} 0$. This explains why the TV distance converges fastest for the zero margin setting (even though that setting does not satisfy stability).

We simulate the algorithm Lin-UCB in the setting of contextual bandits under three different parameter configurations [22]. The "undominated" configuration represents a case where each arm is optimal for some choice of context. In the "dominated" case, one arm is never optimal for any choice of context, and in the "duplicate" case, two arms share the same parameters. In Figure 1, we empirically see that the convergence of the posterior to the representative normal is fastest for the undominated configuration, possibly because in this case, the optimal policy samples each arm with probability bounded away from zero. In the configuration where two arms are duplicate, there may be substantial bias in the estimation of the arm parameters [26], leading the "duplicate" case to have the slowest convergence as seen in Figure 1.

We can also model the Linear Quadratic Regulator (LQR) [24], a common setting used in control theory, using our adaptive linear Gaussian framework. In LQR, we control a state transition of the

form $x_{j+1} = Ax_j + Bu_j + \epsilon_j$ where $A \in \mathbb{R}^{k \times k}, B \in \mathbb{R}^{k \times d}$, $u_j$ are adaptively chosen actions, $\epsilon_j \sim \mathcal{N}(0, \sigma^2 I_k)$ for $j = 1, \ldots, n$, and we aim to estimate the transition matrices $A, B$. To represent this as an instance of Algorithm 1, we serialize the observed state vectors $x_j$ into the sequence of observed outcome variables—that is, we use a trajectory of length $\tilde{n} = nk$ where $\tilde{y}_{(i-1)k+1}, \ldots, \tilde{y}_{ik}$ represents $x_i$. We let the parameter value $\beta$ be $(A \quad B)$ in row major order. Then, the $((i-1)(k+d)+1)$–$(i(k+d))$ indices of the covariate $\tilde{x}_{(j-1)k+i} \in \mathbb{R}^{k(k+d)}$ represent $\begin{pmatrix} x_j \\ u_j \end{pmatrix}$ and all other entries are zero.

We simulate the Noisy Certainty Equivalent Control (NCEC) algorithm on LQR under three different configurations [30]. The "determined" configuration represents a case where the action space has the same dimensionality as the state space and the action transition matrix is full rank. The "stabilizable" case is underdetermined with fewer action dimensions than state dimensions but where the optimal policy allows the system to be stable. The "unstabilizable" case is underdetermined where no policy makes the system stable. In all settings, we empirically see that the BvM TV distance decreases over time, with the convergence rate being the fastest for the unstabilizable case as it has the highest growth rate of $\lambda_{\min}(\mathbf{X}_n^\top \mathbf{X}_n)$. Finally, as we show in Proposition 3 in Appendix D, the NCEC algorithm satisfies condition *(i)* in Theorem 1 so long as a certain stability condition holds (see Assumption 1 in the same appendix section for a precise definition); both our "determined" and "stabilizable" simulation settings satisfy this condition.

We also simulate Thompson Sampling in the two-stage Gaussian batched bandit with two arms, which is the setting analyzed in Theorem 2 of [33]. We compute the TV distance between the posterior and the representative normal in the same way for different values of the margin. As shown in Figure 3, we again see that the average BvM distance is smallest when the margin is zero. We also plot the empirical coverage of the $95\%$ credible interval for the margin, which matches the coverage level except near the zero-margin case, as suggested by [33]. This is an example of a fairly well-behaved sampling process where our BvM result applies but Bayesian credible intervals are still not asymptotically frequentist-valid. Note that this setting satisfies the stability condition if and only if the margin is nonzero. This suggests that the asymptotic frequentist invalidity of Bayesian credible intervals is a local phenomenon around the zero-margin case. Figure 3 supports this analysis and also reveals that in this local region, some parameter values lead to overcoverage and some to undercoverage from a frequentist perspective. Our BvM result may provide an explanation for this, as the Bayesian coverage under a prior containing this local zero-margin region must be correct, meaning the coverage probability aggregated over the erroneous local region should match the coverage level.

Appendix G contains a demonstration of our main results on a simple real-world dataset. Coverage plots for the other simulations are shown in Appendix H.

# 7   Discussion and Future Work

This paper has shown that, for a number of important classes of adaptively collected data, a Bernstein–von Mises theorem applies, linking Bayesian UQ and Wald-type frequentist UQ. This ensures that under extremely mild conditions, Bayesian UQ is asymptotically Bayesian-valid even when the prior is misspecified, and when the stability condition of [21] holds, it also ensures Bayesian UQ is asymptotically frequentist-valid.

One of the surprising takeaways from our work is that when the stability condition of [21] fails, BvM holds but Bayesian UQ is asymptotically frequentist invalid. We note that our work, however, only considers the case when a *fixed, nonrandom* prior is used for Bayesian UQ. This raises the question of whether there is a *data-dependent* way to set the prior so that Bayesian UQ is always asymptotically frequentist valid; we could think of such a method as *empirical* Bayesian UQ.

Another direction of inquiry would be to extend the BvM result to general parametric models beyond just the Gaussian and exponential family cases. However, there are several obstacles to showing the adaptive BvM result in more general parametric models which we discuss in Appendix F.

Note that this paper does not suggest a method of asymptotically frequentist-valid Bayesian inference; in fact, we would like to warn practitioners against using either Wald-type frequentist UQ or Bayesian UQ as if it were asymptotically frequentist-valid.

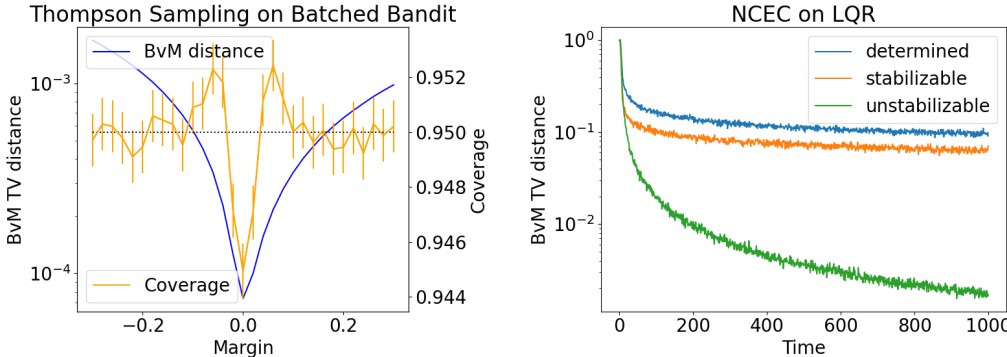

Figure 3: (Left) Average BvM TV distance and empirical coverage of the $95\%$ credible interval for the margin for Thompson Sampling in the two-batch two-arm Gaussian bandit setting with $10^4$ samples per batch. Error bars are $95\%$ confidence intervals over $2 \times 10^5$ replicates. Blacked dotted line is the correct coverage level. $\mathcal{N}(0,1)$ priors are used. (Right) Average TV distance for Noisy Certainty Equivalent Control on LQR [30] under three different parameter configurations. Standard Gaussian priors are used for all arms. TV estimates shown have standard error at most $0.2$ times the TV estimate.

**Declaration of LLM usage:**

ChatGPT-4o was used to create code templates for a Python implementation of the lin-UCB and Stepwise Noisy Certainty Equivalent Control algorithms. The authors revised the templates to ensure correct implementation and modified them to verify the BvM statement.

**Broader Impact:**

The results presented here have the potential to guide the design of safer systems and more reliable hypothesis testing in adaptive experiments. However, it should be noted that the Bernstein–von Mises theorem does not immediately imply frequentist-valid inference as discussed. Thus, Bayesian UQ should *not* in general be treated as frequentist-valid. Instead, our result contributes to a more complete understanding of the differences between frequentist and Bayesian approaches in adaptively collected data.

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

# A  Source Code

The code for this project can be accessed at https://github.com/TheDukeVin/BvM/tree/main. Each figure's data takes under three hours of CPU time to generate.

# B  Main Theorems

## B.1  Proof of Theorem 1

We first write an explicit expression for the posterior. Note that the likelihood can be written as, after removing constant factors that don't depend on $\beta$,

$$
\begin{aligned}
L(\beta; H_n) &= \left(\prod_{j=1}^{n} \Lambda(x_j | H_{j-1})\right) \left(\prod_{j=1}^{n} P(y_j | x_j, \beta)\right) \\
&\propto \prod_{j=1}^{n} P(y_j | x_j, \beta) \propto \prod_{j=1}^{n} \exp\left(-\frac{(y_j - x_j^\top \beta)^2}{2\sigma^2}\right) \\
&= \prod_{j=1}^{n} \exp\left(-\frac{(y_j - \hat{\beta}_n^\top x_j)^2}{2\sigma^2} - \frac{(\hat{\beta}_n^\top x_j - \beta^\top x_j)^2}{2\sigma^2}\right) \\
&\propto \exp\left(-\frac{1}{2\sigma^2}(\hat{\beta}_n - \beta)^\top \mathbf{X}_n^\top \mathbf{X}_n (\hat{\beta}_n - \beta)\right).
\end{aligned}
$$

Thus, by Bayes' rule, the posterior can be written as

$$
\pi(\beta | H_n) \propto \pi(\beta) \exp\left(-\frac{1}{2\sigma^2}(\hat{\beta}_n - \beta)^\top \mathbf{X}_n^\top \mathbf{X}_n (\hat{\beta}_n - \beta)\right).
$$

Let $d(H_n) = \|\pi(\beta | H_n) - \mathcal{N}(\hat{\beta}_n, \sigma^2(\mathbf{X}_n^\top \mathbf{X}_n)^{-1})\|_{\mathrm{TV}}$. By assumption, $\pi(\beta_0) > 0$, so using Lemma 2, we have

$$
\begin{aligned}
d(H_n) &\leq \int \left(\frac{\pi(\beta)}{\pi(\beta_0)} \frac{1}{\sqrt{(2\pi\sigma^2)^p |\mathbf{X}_n^\top \mathbf{X}_n|^{-1}}} \exp\left(-\frac{1}{2\sigma^2}(\beta - \hat{\beta}_n)^\top \mathbf{X}_n^\top \mathbf{X}_n (\beta - \hat{\beta}_n)\right)\right. \\
&\qquad \left. - \frac{1}{\sqrt{(2\pi\sigma^2)^p |\mathbf{X}_n^\top \mathbf{X}_n|^{-1}}} \exp\left(-\frac{1}{2\sigma^2}(\beta - \hat{\beta}_n)^\top \mathbf{X}_n^\top \mathbf{X}_n (\beta - \hat{\beta}_n)\right)\right)_+ d\beta \\
&= \int \left(\frac{\pi(\beta)}{\pi(\beta_0)} - 1\right)_+ \frac{1}{\sqrt{(2\pi\sigma^2)^p |\mathbf{X}_n^\top \mathbf{X}_n|^{-1}}} \exp\left(-\frac{1}{2\sigma^2}(\beta - \hat{\beta}_n)^\top \mathbf{X}_n^\top \mathbf{X}_n (\beta - \hat{\beta}_n)\right) d\beta. \\
&= \mathbb{E}\left[\left(\frac{\pi(\kappa_n)}{\pi(\beta_0)} - 1\right)_+ \Big| H_n\right]
\end{aligned}
$$

where $\kappa_n$ is a random variable marginally distributed as $\kappa_n | H_n \sim \mathcal{N}(\hat{\beta}_n, \sigma^2(\mathbf{X}_n^\top \mathbf{X}_n)^{-1})$. Let $c_n$ be a sequence of random variables satisfying $\frac{c_n}{\log \lambda_{\max}(\mathbf{X}_n^\top \mathbf{X}_n)} \overset{p}{\to} \infty$ and $\frac{c_n}{\lambda_{\min}(\mathbf{X}_n^\top \mathbf{X}_n)} \overset{p}{\to} 0$, which must exist by condition *(i)*, i.e. we can let $c_n = \sqrt{\lambda_{\min}(\mathbf{X}_n^\top \mathbf{X}_n) \log \lambda_{\max}(\mathbf{X}_n^\top \mathbf{X}_n)}$. Then, to show $d(H_n) \overset{p}{\to} 0$, we use a common trick when showing BvM-style results [29, 11, 18]—that is, we will partition the expectation into parts inside and outside a local ellipsoid. We will show the following two statements.

$$\mathbb{E}\left[\left(\frac{\pi(\kappa_n)}{\pi(\beta_0)}-1\right)_+ 1[\|(\mathbf{X}_n^\top\mathbf{X}_n)^{1/2}(\kappa_n-\hat{\beta}_n)\|^2 \le c_n]\Big|H_n\right] \xrightarrow{p} 0 \tag{1}$$

$$\mathbb{E}\left[\left(\frac{\pi(\kappa_n)}{\pi(\beta_0)}-1\right)_+ 1[\|(\mathbf{X}_n^\top\mathbf{X}_n)^{1/2}(\kappa_n-\hat{\beta}_n)\|^2 > c_n]\Big|H_n\right] \xrightarrow{p} 0 \tag{2}$$

To show (1), note that we have

$$\mathbb{E}\left[\left(\frac{\pi(\kappa_n)}{\pi(\beta_0)}-1\right)_+ 1[\|(\mathbf{X}_n^\top\mathbf{X}_n)^{1/2}(\kappa_n-\hat{\beta}_n)\|^2 \le c_n]\Big|H_n\right]$$

$$\le \frac{1}{\pi(\beta_0)}\mathbb{E}\left[(\pi(\kappa_n)-\pi(\beta_0))_+ 1[\|(\kappa_n-\hat{\beta}_n)\|^2 \le \frac{c_n}{\lambda_{\min}(\mathbf{X}_n^\top\mathbf{X}_n)}]\Big|H_n\right]$$

$$\le \frac{1}{\pi(\beta_0)}\sup_{\|\kappa_n-\hat{\beta}_n\| \le \frac{c_n^{1/2}}{\lambda_{\min}(\mathbf{X}_n^\top\mathbf{X}_n)^{1/2}}} (\pi(\kappa_n)-\pi(\beta_0))_+$$

$$\le \frac{1}{\pi(\beta_0)}\sup_{\|\kappa_n-\beta_0\| \le \frac{c_n^{1/2}}{\lambda_{\min}(\mathbf{X}_n^\top\mathbf{X}_n)^{1/2}}+\|\hat{\beta}_n-\beta_0\|} (\pi(\kappa_n)-\pi(\beta_0))_+$$

Lai and Wei showed that $\hat{\beta}_n$ is consistent if condition *(i)* is satisfied [21]. Thus, we have $\frac{c_n^{1/2}}{\lambda_{\min}(\mathbf{X}_n^\top\mathbf{X}_n)^{1/2}} + \|\hat{\beta}_n - \beta_0\| \xrightarrow{p} 0$, meaning the above expression does indeed converge to zero in probability if $\pi$ is continuous at $\beta_0$.

To show (2), note that by the triangle inequality, we have

$$\mathbb{E}\left[\left(\frac{\pi(\kappa_n)}{\pi(\beta_0)}-1\right)_+ 1[\|(\mathbf{X}_n^\top\mathbf{X}_n)^{1/2}(\kappa_n-\hat{\beta}_n)\|^2 > c_n]\Big|H_n\right]$$

$$\le \frac{1}{\pi(\beta_0)}\mathbb{E}\left[\pi(\kappa_n)1[\|(\mathbf{X}_n^\top\mathbf{X}_n)^{1/2}(\kappa_n-\hat{\beta}_n)\|^2 > c_n]\Big|H_n\right]$$

$$+ \mathbb{E}\left[1[\|(\mathbf{X}_n^\top\mathbf{X}_n)^{1/2}(\kappa_n-\hat{\beta}_n)\|^2 > c_n]\Big|H_n\right]$$

The first term can then be bounded as

$$\frac{1}{\pi(\beta_0)}\mathbb{E}\left[\pi(\kappa_n)1[\|(\mathbf{X}_n^\top\mathbf{X}_n)^{1/2}(\kappa_n-\hat{\beta}_n)\|^2 > c_n]\Big|H_n\right]$$

$$= \frac{1}{\pi(\beta_0)}\int \pi(\kappa_n)1[\|(\mathbf{X}_n^\top\mathbf{X}_n)^{1/2}(\kappa_n-\hat{\beta}_n)\|^2 > c_n]P(\kappa_n|H_n)d\kappa_n$$

$$\le \frac{1}{\pi(\beta_0)}\int \pi(\kappa_n)1[\|(\mathbf{X}_n^\top\mathbf{X}_n)^{1/2}(\kappa_n-\hat{\beta}_n)\|^2 > c_n]\frac{1}{\sqrt{(2\pi\sigma^2)^p|\mathbf{X}_n^\top\mathbf{X}_n|^{-1}}}\exp(-\frac{c_n}{2\sigma^2})d\kappa_n$$

$$\le \frac{(2\pi\sigma^2)^{-p/2}}{\pi(\beta_0)}\exp(-\frac{c_n}{2\sigma^2})\lambda_{\max}(\mathbf{X}_n^\top\mathbf{X}_n)^{p/2}\int \pi(\kappa_n)1[\|(\mathbf{X}_n^\top\mathbf{X}_n)^{1/2}(\kappa_n-\hat{\beta}_n)\|^2 > c_n]d\kappa_n$$

$$\le \frac{(2\pi\sigma^2)^{-p/2}}{\pi(\beta_0)}\exp(-\frac{c_n}{2\sigma^2}+\frac{p}{2}\log\lambda_{\max}(\mathbf{X}_n^\top\mathbf{X}_n))\int \pi(\kappa_n)d\kappa_n$$

$$= \frac{(2\pi\sigma^2)^{-p/2}}{\pi(\beta_0)}\exp(-\frac{c_n}{2\sigma^2}+\frac{p}{2}\log\lambda_{\max}(\mathbf{X}_n^\top\mathbf{X}_n))$$

If $\frac{c_n}{\log \lambda_{\max}(\mathbf{X}_n^\top \mathbf{X}_n)} \overset{P}{\to} \infty$, the above expression converges to zero in probability. To compute the second term, note that $(\mathbf{X}_n^\top \mathbf{X}_n)^{1/2}(\kappa_n - \hat{\beta}_n)|H_n$ is distributed as $\mathcal{N}(0, \sigma^2 I_p)$. Thus, if $c_n \overset{P}{\to} \infty$, then $\mathbb{E}\left[1[\|(\mathbf{X}_n^\top \mathbf{X}_n)^{1/2}(\kappa_n - \hat{\beta}_n)\|^2 > c_n]\Big|H_n\right] \overset{P}{\to} 0$. We have shown that each term (1) and (2) in the decomposition of the upper bound for $d(H_n)$ converges to zero in probability, meaning $d(H_n)$ converges to zero in probability as well.

## B.2   Proof of Theorem 2

As in the proof of Theorem 1, the posterior distribution can be expressed as

$$\pi(\beta|H_{m_n}^n) \propto \pi(\beta) \exp(-\frac{1}{2\sigma^2}(\hat{\beta}_n - \beta)^\top \mathbf{X}_n^\top \mathbf{X}_n(\hat{\beta}_n - \beta)).$$

Letting $d(H_{m_n}^n) = \|\pi(\beta|H_{m_n}^n) - \mathcal{N}(\hat{\beta}_n, \sigma^2(\mathbf{X}_n^\top \mathbf{X}_n)^{-1})\|_{\text{TV}}$, we also have

$$d(H_{m_n}^n) \le \mathbb{E}\left[\left(\frac{\pi(\kappa_n)}{\pi(\beta_0)} - 1\right)_+ \Big| H_{m_n}^n\right]$$

where $\kappa_n|H_{m_n}^n \sim \mathcal{N}(\hat{\beta}_n, \sigma^2(\mathbf{X}_n^\top \mathbf{X}_n)^{-1})$. By Lemma 1, we have that $\hat{\beta}_n$ is consistent. If $\lambda_{\min}(\mathbf{X}_n^\top \mathbf{X}_n) \overset{P}{\to} \infty$, then $\kappa_n \overset{P}{\to} \beta_0$ as we can write $\kappa_n = \beta_0 + (\hat{\beta}_n - \beta_0) + Z_n$ where $Z_n|H_{m_n}^n \sim \mathcal{N}(0, \sigma^2(\mathbf{X}_n^\top \mathbf{X}_n)^{-1})$ and both the terms $\hat{\beta}_n - \beta_0$ and $Z_n$ converge to zero in probability. If $\pi$ is continuous and bounded, by Vitali's convergence theorem,

$$\mathbb{E}[d(H_{m_n}^n)] \le \mathbb{E}\left[\left(\frac{\pi(\kappa_n)}{\pi(\beta_0)} - 1\right)_+\right] \to 0,$$

meaning $d(H_{m_n}^n) \overset{P}{\to} 0$.

## B.3   Proof of Theorem 3

Suppose $Y_{i,(1)}, Y_{i,(2)}, \ldots$ are the serialized arm pulls from arm $i$. Let $\bar{Y}_{i,(N)} = \frac{1}{N}\sum_{j=1}^N y_{i,(j)}$ be the sequence of sample means. By Lemma 3, we have

$$\frac{\bar{Y}_{i,(N)} - \mu}{\sigma} = o(N^{-1/2+\epsilon}) \text{ almost surely as } N \to \infty. \tag{3}$$

for some small $\epsilon$, say $\epsilon = 0.01$. Since $\bar{Y}_{n,i} = \bar{Y}_{i,(N_{n,i})}$, we have that $\bar{Y}_{n,i} - \mu_i = O_p(N_{n,i}^{-1/2+\epsilon})$. Thus, $\hat{\beta}_{n,i} - \beta_{0,i} = O_p(N_{n,i}^{-1/2+\epsilon})$, meaning $\hat{\beta}_{n,i}$ is consistent for $\beta_{0,i}$.

We prove the result in three steps by truncating both the posterior and normal distribution to the set $C_n = \{\beta : \forall i, |\beta_i - \beta_{0,i}| \le c_{n,i}\}$ for some sequence of random variables $c_{n,i}$ such that $N_{n,i}^{1/3} c_{n,i} \overset{P}{\to} 0$ and $N_{n,i}^{1/2-\epsilon} c_{n,i} \overset{P}{\to} \infty$ as $n \to \infty$. We then show that

$$\|\pi(\beta|H_n) - \pi^{C_n}(\beta|H_n)\|_{\text{TV}} \overset{P}{\to} 0 \tag{4}$$

$$\|\pi^{C_n}(\beta|H_n) - \mathcal{N}^{C_n}(\beta_0, I_n^{-1})\|_{\text{TV}} \overset{P}{\to} 0 \tag{5}$$

$$\|\mathcal{N}^{C_n}(\beta_0, I_n^{-1}) - \mathcal{N}(\beta_0, I_n^{-1})\|_{\text{TV}} \overset{P}{\to} 0. \tag{6}$$

We first show (5). The likelihood function can be calculated as

$$L(\beta; H_n) \propto \prod_{i=1}^p \exp(N_{n,i}(\bar{Y}_{n,i}\beta_i - b(\beta_i))).$$

By Taylor's Theorem, for any $\beta \in C_n$, we have

$$L(\beta; H_n) \propto \exp\left(\sum_{i=1}^{p} N_{n,i}[\bar{Y}_{n,i}\beta_i - b(\beta_{0,i}) - b'(\beta_{0,i})(\beta_i - \beta_{0,i})\right.$$
$$\left. - \frac{1}{2}b''(\beta_{0,i})(\beta_i - \beta_{0,i})^2 - \frac{1}{6}b'''(\beta_{n,\beta,i}^*)(\beta_i - \beta_{0,i})^3]\right)$$

for some $\beta_{n,\beta}^* \in C_n$. However, if $b'''$ is bounded on a neighborhood of $\beta_{0,i}$ for each $i$ and $N_{n,i}^{1/3}c_{n,i} \xrightarrow{p} 0$, we have that $\sup_{\beta \in C_n} N_{n,i}b'''(\beta_{n,\beta,i}^*)(\beta_i - \beta_{0,i})^3 \xrightarrow{p} 0$. Thus, we can write

$$L(\beta; H_n) \propto \exp\left(\sum_{i=1}^{p} N_{n,i}[\bar{Y}_{n,i}\beta_i - b(\beta_{0,i}) - b'(\beta_{0,i})(\beta_i - \beta_{0,i}) - \frac{1}{2}b''(\beta_{0,i})(\beta_i - \beta_{0,i})^2] + o_p(1)\right)$$

$$\propto \exp\left(\sum_{i=1}^{p} N_{n,i}[-\frac{1}{2}b''(\beta_{0,i})\beta_i^2 + (\bar{Y}_{n,i} - b'(\beta_{0,i}) + b''(\beta_{0,i})\beta_{0,i})\beta_i]\right)(1 + o_p(1))$$

$$= \exp\left(\sum_{i=1}^{p} N_{n,i}[-\frac{1}{2}b''(\beta_{0,i})\beta_i^2 + b''(\beta_{0,i})\hat{\beta}_{n,i}\beta_i]\right)(1 + o_p(1))$$

$$\propto \exp\left(-\frac{1}{2}\sum_{i=1}^{p} N_{n,i}b''(\beta_{0,i})(\beta_i - \hat{\beta}_{n,i})^2\right)(1 + o_p(1)) \tag{7}$$

$$= \exp(-\frac{1}{2}(\beta - \hat{\beta}_n)^\top I_n(\beta - \hat{\beta}_n))(1 + o_p(1))$$

Let the density of the truncated normal distribution $\mathcal{N}^{C_n}(\hat{\beta}_{n,i}, I_n^{-1})$ be expressed as

$$A_n 1_{\beta \in C_n} \exp(-\frac{1}{2}(\beta - \hat{\beta}_{n,i})^\top I_n(\beta - \hat{\beta}_{n,i})).$$

where $A_n$ is chosen such that the density is suitably normalized. Let $d_n(H_n) = \|\pi^{C_n}(\beta|H_n) - \mathcal{N}^{C_n}(\beta_0, I_n^{-1})\|_{\mathrm{TV}}$. By Lemma 2, we have

$$d_n(H_n) \leq \int \left(A_n 1_{\beta \in C_n} \frac{\pi(\beta)L(\beta; H_n)}{\pi(\hat{\beta}_n)L(\hat{\beta}_n; H_n)} - A_n 1_{\beta \in C_n} \exp(-\frac{1}{2}(\beta - \hat{\beta}_n)^\top I_n(\beta - \hat{\beta}_n))\right)_+ d\beta$$

$$\leq \int A_n 1_{\beta \in C_n} \left(\frac{\pi(\beta)}{\pi(\hat{\beta}_n)} \exp(-\frac{1}{2}(\beta - \hat{\beta}_n)^\top I_n(\beta - \hat{\beta}_n))(1 + \bar{o}_p(1))\right.$$

$$\left. - \exp(-\frac{1}{2}(\beta - \hat{\beta}_n)^\top I_n(\beta - \hat{\beta}_n))\right)_+ d\beta$$

where we say a sequence of random variables $W_n(\beta)$ indexed by $\beta$ is $\bar{o}_p(1)$ if $\sup_{\beta \in C_n} W_n(\beta) \xrightarrow{p} 0$. Thus, we have

$$d_n(H_n) \leq \int \left(\left(\frac{\pi(\beta)}{\pi(\hat{\beta}_n)} - 1 + \bar{o}_p(1)\right) A_n 1_{\beta \in C_n} \exp(-\frac{1}{2}(\beta - \hat{\beta}_n)^\top I_n(\beta - \hat{\beta}_n))\right)_+ d\beta.$$

$$= \mathbb{E}_{\eta_n \sim \mathcal{N}^{C_n}(\hat{\beta}_{n,i}, I_n^{-1})}\left(\frac{\pi(\eta_n)}{\pi(\hat{\beta}_n)} - 1 + \bar{o}_p(1)\right)_+$$

$$\leq \left(\frac{\sup_{\beta \in C_n} \pi(\beta)}{\inf_{\beta \in C_n} \pi(\beta)} - 1 + \bar{o}_p(1)\right)_+$$

Then, since $c_{n,i} \xrightarrow{p} 0$ for each $i$ and $\pi$ is continuous, we know $\frac{\sup_{\beta \in C_n} \pi(\beta)}{\inf_{\beta \in C_n} \pi(\beta)} \xrightarrow{p} 1$, meaning the above expression indeed converges to 0 in probability. Next, we show (4). Note that

$$
\begin{aligned}
\|\pi(\beta|H_n) - \pi^{C_n}(\beta|H_n)\|_{\mathrm{TV}} &= P_\pi(\beta \in C_n^C | H_n) \\
&= \frac{\int_{C_n^C} \pi(\beta) L(\beta; H_n) d\beta}{\int_{\mathbb{R}^p} \pi(\beta) L(\beta; H_n) d\beta} \\
&\leq \frac{\pi^{\max}}{\pi_n^{\min}} \frac{\int_{C_n^C} L(\beta; H_n) d\beta}{\int_{C_n} L(\beta; H_n) d\beta}
\end{aligned}
$$

where $\pi^{\max} = \sup_\beta \pi(\beta)$ and $\pi_n^{\min} = \inf_{\beta \in C_n} \pi(\beta)$. Note that if $\pi$ is continuous and positive at $\beta_0$, then $\pi_n^{\min}$ converges to some positive constant. Then, since the ancillary function $b$ of the exponential family is convex, the likelihood function is log-concave in $\beta$, meaning we have

$$
\begin{aligned}
\frac{\int_{C_n^C} L(\beta; H_n) d\beta}{\int_{C_n} L(\beta; H_n) d\beta} &\leq \sup_{v \in \partial C_n} \frac{\int_1^\infty L(\beta_0 + (v - \beta_0)t; H_n) t^{p-1} dt}{\int_0^1 L(\beta_0 + (v - \beta_0)t; H_n) t^{p-1} dt} \\
&\leq \sup_{v \in \partial C_n} \frac{\int_1^\infty \exp(\ell_n(\beta_0) + (\ell_n(v) - \ell_n(\beta_0))t) t^{p-1} dt}{\int_0^1 \exp(\ell_n(\beta_0) + (\ell_n(v) - \ell_n(\beta_0))t) t^{p-1} dt} \\
&= \sup_{v \in \partial C_n} \frac{\int_{\ell_n(\beta_0) - \ell_n(v)}^\infty \exp(-x) x^{p-1} dx}{\int_0^{\ell_n(\beta_0) - \ell_n(v)} \exp(-x) x^{p-1} dx} \\
&= \frac{\int_{a_n}^\infty \exp(-x) x^{p-1} dx}{\int_0^{a_n} \exp(-x) x^{p-1} dx}
\end{aligned}
$$

where $a_n = \inf_{v \in \partial C_n} \ell_n(\beta_0) - \ell_n(v)$. Note that the function $a \mapsto \frac{\int_a^\infty \exp(-x) x^{p-1} dx}{\int_0^a \exp(-x) x^{p-1} dx}$ goes to zero as $a \to \infty$. Thus, it suffices to show that $a_n \xrightarrow{p} \infty$. To do this, note that by expression 7, we have

$$
a_n = \inf_{v \in \partial C_n} -\frac{1}{2} \sum_{i=1}^p N_{n,i} b''(\beta_{0,i})[(\beta_{0,i} - \hat{\beta}_{n,i})^2 - (v_i - \hat{\beta}_{n,i})^2] + \bar{o}_p(1).
$$

Since $|v_i - \hat{\beta}_{n,i}| \geq \|v_i - \beta_{0,i}| - |\beta_{0,i} - \hat{\beta}_{n,i}\| = |c_{n,i} - |\beta_{0,i} - \hat{\beta}_{n,i}\|$, we have

$$
\begin{aligned}
a_n &\geq \inf_{v \in \partial C_n} -\frac{1}{2} \sum_{i=1}^p N_{n,i} b''(\beta_{0,i})[|\beta_{0,i} - \hat{\beta}_{n,i}|^2 - (c_{n,i} - |\beta_{0,i} - \hat{\beta}_{n,i}|)^2] + \bar{o}_p(1) \\
&= \inf_{v \in \partial C_n} -\frac{1}{2} \sum_{i=1}^p N_{n,i} b''(\beta_{0,i})[|\beta_{0,i} - \hat{\beta}_{n,i}|^2 - c_{n,i}^2 + 2c_{n,i}|\beta_{0,i} - \hat{\beta}_{n,i}| - |\beta_{0,i} - \hat{\beta}_{n,i}|^2] + \bar{o}_p(1) \\
&= \inf_{v \in \partial C_n} \frac{1}{2} \sum_{i=1}^p c_{n,i}^2 N_{n,i} b''(\beta_{0,i})[1 - \frac{2|\beta_{0,i} - \hat{\beta}_{n,i}|}{c_{n,i}}] + \bar{o}_p(1)
\end{aligned}
$$

By the definition of $c_{n,i}$ as well as Equation (3), we know $c_{n,i}^2 N_{n,i} \xrightarrow{p} \infty$ and $\frac{\beta_{0,i} - \hat{\beta}_{n,i}}{c_{n,i}} \xrightarrow{p} 0$. Thus, we indeed have $a_n \xrightarrow{p} \infty$. Lastly, we show (6). We have

$$
\begin{aligned}
\|\mathcal{N}(\beta_0, I_n^{-1}) - \mathcal{N}^{C_n}(\beta_0, I_n^{-1})\|_{\mathrm{TV}} &= P_{X \sim \mathcal{N}(\beta_0, I_n^{-1})}(\exists i, |X_i - \beta_{0,i}| > c_{n,i}) \\
&= P_{X \sim \mathcal{N}(0, I_{p \times p})}(\exists i, |X_i| > c_{n,i} N_{n,i}^{1/2} b''(\beta_{0,i})^{1/2}).
\end{aligned}
$$

If $c_{n,i} N_{n,i}^{1/2} \xrightarrow{p} \infty$, then the above expression indeed coverges to zero in probability.

## B.4 Proof of Theorem 4

We only need to establish the consistency of $\hat{\beta}_n$, after which the result follows by a similar argument to the proof of Theorem 2. Consistency can be shown from condition (ii) by Theorem 1 of [21].

## C  Technical Lemmas

**Proof of Lemma 1:**  We translate this directly to the bandit setting, after which this result follows from a similar version of Theorem 2.2 of [13]. For each $i = 1, \ldots, p$, let $\mu_i = \beta_i$ be the $i$-th coordinate of $\beta$ which represents the true mean of arm $i$. Let $N_{i,n} = (\mathbf{X}_n)_{ii}$ be the number of times arm $i$ is pulled in the trajectory $H^n_{m_n}$. We serialize all arm pulls from each arm $i$—that is, suppose $y^n_{i,(1)}, y^n_{i,(2)}, \ldots \overset{ind}{\sim} \mathcal{N}(\mu_i, \sigma^2)$. Then, when drawing $y^n_j | x^n_j \sim \mathcal{N}(\mu_{x_j}, \sigma^2)$, we look up the next unused sample in the serialized sequence $y^n_{x_j,(1)}, \ldots$ and use that as our observation.

Under this formulation of the data-generating process, the sample mean $\hat{\beta}_{n,i}$ is simply the mean of the first $N_{i,n}$ samples from the serialized sequence $y^n_{i,(1)}, \ldots$. Note that the serialized sample means $\bar{y}^n_{i,t} = \frac{1}{t} \sum_{j=1}^t y^n_{i,(j)}$ satisfy $\bar{y}^n_{i,t} \overset{a.s.}{\to} \mu_i$ as $t \to \infty$ for each $i, n$ by the strong law of large numbers. Then, Proposition 1 implies that for each $i, n$, there exists some function $\epsilon^n_i(\cdot)$ satisfying $\lim_{B \to \infty} \epsilon^n_i(B) = 0$ such that for any $B$,

$$\mathbb{P}[N \geq B] \geq 1 - \frac{1}{B} \implies \mathbb{P}[|\bar{y}^n_{i,(N)} - \mu_i| \leq \epsilon^n_i(B)] \geq 1 - \epsilon^n_i(B)$$

Crucially, note that the functions $\epsilon^n_i(\cdot)$ can be chosen such that $\epsilon^n_i(\cdot)$ does not depend on $n$. This is because the distribution of serialized samples $\{y^n_{i,(j)}\}_{i \in \{1,\ldots,p\}, j \geq 1}$ for each trajectory is identical, and in Proposition 1, the bound translation function $\epsilon$ is chosen solely based on the distribution of the almost-surely convergent sequence. Finally, by condition *(ii)*, we indeed have that $N_{i,n} \overset{p}{\to} \infty$ for each $i$, meaning the condition $\mathbb{P}[N_{i,n} \geq B_n] \geq 1 - \frac{1}{B_n}$ is indeed satisfied for some sequence $B_n \to \infty$ - to see this, note that letting $B_n = \sup\{B : \mathbb{P}[N_{i,n} \geq B] \geq 1 - \frac{1}{B}\}$, we must have $B_n \to \infty$ as otherwise there would be some constant $C$ where $\mathbb{P}[N_{i,n} \leq C] > \frac{1}{C}$ for infinitely many $n$ which would contradict the statement that $N_{i,n}$ diverges in probability. Thus, we have $\mathbb{P}[|\bar{y}^n_{i,(N_{i,n})} - \mu_i| \leq \epsilon^n_i(B_n)] \geq 1 - \epsilon^n_i(B_n)$ for each $n$ meaning the sample mean $\hat{\beta}_{n,i} = \bar{y}^n_{i,(N_{i,n})}$ is indeed consistent for $\mu_i$.

**Proposition 1.** *(Bound translation) Suppose $Y_n$ is a sequence of random variables such that $Y_n \overset{a.s.}{\to} Y$ for some real number $Y$. Then there exists a function $\epsilon(B)$ with $\epsilon(B) \to 0$ as $B \to \infty$ such that for any random variable $N$ and positive integer $B$, we have*

$$\mathbb{P}[N \geq B] \geq 1 - \frac{1}{B} \implies \mathbb{P}[|Y_N - Y| \leq \epsilon(B)] \geq 1 - \epsilon(B).$$

*Proof.* Note that it is sufficient to show that for any fixed $\eta$, there is some function $\epsilon_\eta(B)$ such that $\lim_{B \to \infty} \epsilon_\eta(B) = 0$ and

$$\mathbb{P}[N \geq B] \geq 1 - \frac{1}{B} \implies \mathbb{P}[|Y_N - Y| \leq \eta] \geq 1 - \epsilon_\eta(B).$$

To show this, we simply let $\epsilon_\eta(B) = \frac{1}{B} + \mathbb{P}[\exists n \geq B, |Y_n - Y| > \eta]$. If $Y_n$ converges almost surely to $Y$, then $\lim_{B \to \infty} \epsilon_\eta(B) = 0$. Also, we indeed have

$$\mathbb{P}[|Y_N - Y| \leq \eta] \geq \mathbb{P}[N \geq B \text{ and } \forall n \geq B, |Y_n - Y| \leq \eta] \geq 1 - \epsilon_\eta(B).$$

Since we can do this for any $\eta$, the statement of the result is indeed true. $\qquad\square$

**Lemma 2.** *Let $P(\cdot)$ and $Q(\cdot)$ be continuous distributions and $c \in \mathbb{R}$ be any constant. Then, $\|P - Q\|_{\mathrm{TV}} = \frac{1}{2} \int (P(x) - Q(x))_+ dx \leq \int (cP(x) - Q(x))_+ dx$.*

*Proof.* Let $A = \{x : P(x) > Q(x)\}$. Let $a_1 = \int_A P(x) - Q(x)dx$ and $a_2 = \int_{A^C} Q(x) - P(x)dx$. Then, note that

$$\|P - Q\|_{\text{TV}} = \frac{1}{2}\int (P(x) - Q(x))_+ dx = \frac{1}{2}a_1 + \frac{1}{2}a_2$$

and

$$0 = \int P(x) - Q(x)dx = a_1 - a_2$$

Solving for $a_1$ and $a_2$, we get

$$a_1 = a_2 = \|P - Q\|_{\text{TV}}.$$

To show the desired result, if $c \geq 1$, we have

$$\|P - Q\|_{\text{TV}} = a_1 \leq \int_A cP(x) - Q(x)dx \leq \int (cP(x) - Q(x))_+ dx.$$

Similarly, if $c \leq 1$, then

$$\|P - Q\|_{\text{TV}} = a_2 \leq \int_{A^C} Q(x) - cP(x)dx \leq \int (cP(x) - Q(x))_+ dx.$$

$\square$

**Lemma 3.** *Let $X_1, X_2, \ldots$ be i.i.d. samples from a distribution $P$ with mean $\mu$ and finite variance $\sigma^2$. Let $\hat{\mu}_n = \frac{1}{n}\sum_{i=1}^n X_i$. Then,*

$$n^{1/2-\epsilon}(\hat{\mu}_n - \mu) \overset{a.s.}{\to} 0.$$

*for any $\epsilon > 0$.*

*Proof.* By the law of iterated logarithms, we have

$$\limsup_{n\to\infty} \frac{\sqrt{n}|\hat{\mu}_n - \mu|}{\sqrt{2\sigma^2 \log\log n}} = 1$$

with probability 1. Then, we have

$$\limsup_{n\to\infty} n^{1/2-\epsilon}|\hat{\mu}_n - \mu| \leq \left(\limsup_{n\to\infty} \frac{\sqrt{n}|\hat{\mu}_n - \mu|}{\sqrt{2\sigma^2 \log\log n}}\right)\left(\limsup_{n\to\infty} \frac{\sqrt{2\sigma^2 \log\log n}}{n^\epsilon}\right) = 0$$

with probability 1. $\square$

## D   Auxilliary results

**Proposition 2.** *Let $x_1$ be deterministic and take any real value. Let the adaptive sampling procedure simply deterministically set $x_i = y_{i-1}$ for $i > 1$. Then, if $\beta = 1$, the MLE is asymptotically non-Normal. However, condition (i) of Theorem 1 is satisfied by this sampling design.*

*Proof.* The asymptotic non-Normality is established by Lai and Wei in [21, Example 3]. The second part of condition *(i)* is satisfied because the covariates are 1-dimensional and the first part is satisfied because $\sum_{i=1}^n x_i^2 \overset{p}{\to} \infty$, which is a direct consequence of [21, Equation (4.9)]. $\square$

We now recall an assumption from [30]:

**Assumption 1** (Stability, [30, Assumption 1]). *There exists a matrix $K_0$ for which the maximum absolute eigenvalue of $A + BK_0$ is less than 1.*

**Proposition 3.** *Suppose we run the stepwise NCEC algorithm (Algorithm 1 in [30]) on an instance of LQR satisfying Assumption 1 using any configuration of the hyperparameters $\tau^2 > 0$, $\beta \in [1/2, 1)$ and $\alpha > 0$ permitted therein. Then, this satisfies condition* (i) *of Theorem 1.*

*Proof.* First, observe that our Gram matrix $\tilde{X}^\top \tilde{X}$ is more simply expressed as $I_{k \times k} \otimes \boldsymbol{G}_n$ where $\otimes$ denotes Kronecker product and $\boldsymbol{G}_n := \sum_{i=1}^n \begin{bmatrix} x_i \\ u_i \end{bmatrix} \begin{bmatrix} x_i \\ u_i \end{bmatrix}^\top$ denotes the Gram matrix. Because the largest and smallest eigenvalues of $I_{k \times k} \otimes \boldsymbol{G}_n$ are the same as that of $\boldsymbol{G}_n$ it suffices to study the spectrum of the latter. Theorem 3 of [30] shows, in the stabilizable regime, that

$$D_n^{-1} \boldsymbol{G}_n \left( D_n^{-1} \right)^\top \xrightarrow{p} I_{k+d},$$

where

$$D_n = n^{\beta/2} \log^{\alpha/2}(n) \begin{bmatrix} I_k & 0 \\ K & I_d \end{bmatrix} \begin{bmatrix} C_n^{1/2} & 0 \\ 0 & \sqrt{\tau^2/\beta} I_d \end{bmatrix}, \tag{8}$$

$$C_n = n^{1-\beta} \log^{-\alpha}(n) \sum_{p=0}^\infty (A+BK)^p ((A+BK)^p)^\top \sigma^2 + \frac{\tau^2}{\beta} \sum_{q=0}^\infty (A+BK)^q BB^\top ((A+BK)^q)^\top, \tag{9}$$

and $K$ is a matrix that does not depend on $n$ (see equation (3) of [30] for a precise definition).

from which it follows that

$$\left\| \boldsymbol{G}_n - D_n D_n^\top \right\| = o_p(1). \tag{10}$$

Below, we show that $\lambda_{\max} \left( D_n D_n^\top \right) \le O(n)$ and $\lambda_{\min} \left( D_n D_n^\top \right) \ge \Omega(n^\beta \log^\alpha(n))$. Because Equation (10) implies both that $\lambda_{\min}(\boldsymbol{G}_n) - \lambda_{\min} \left( D_n D_n^\top \right) = o_p(1)$ and $\lambda_{\min}(\boldsymbol{G}_n) - \lambda_{\min} \left( D_n D_n^\top \right) = o_p(1)$, we indeed have that condition *(i)* of Theorem 1 is met: $\log \left( \lambda_{\max}(\boldsymbol{G}_n) \right) = o_p(\lambda_{\min}(\boldsymbol{G}_n))$ and $\lambda_{\min}(\boldsymbol{G}_n) \xrightarrow{p} \infty$.

**Upper bound on $\lambda_{\max}(D_n D_n^\top)$** By submultiplicativity of the operator norm, we have, using equation (8), that the largest eigenvalue of $D_n D_n^\top$ is at most

$$n^\beta \log^\alpha(n) \left\| \begin{bmatrix} I_k & 0 \\ K & I_d \end{bmatrix} \right\|^2 \left\| \begin{bmatrix} C_n^{1/2} & 0 \\ 0 & \sqrt{\tau^2/\beta} I_d \end{bmatrix} \right\|^2$$

The middle term is of constant order and therefore it suffices to upper-bound the last term. Because $\left\| \begin{bmatrix} C_n^{1/2} & 0 \\ 0 & \sqrt{\tau^2/\beta} I_d \end{bmatrix} \right\| \le \max \left( \|C_n^{1/2}\|, \|\sqrt{\tau^2/\beta} I_d\| \right)$ and since $\|\sqrt{\tau^2/\beta} I_d\|$ is again of constant order it suffices to simply bound $\|C_n^{1/2}\|$. Using equation (9) and triangle inequality, this is at most $O(n^{(1-\beta)/2} \log^{-\alpha/2}(n))$. Consequently, the maximum eigenvalue of $D_n D_n^\top$ is $O(n)$.

**Lower bound on $\lambda_{\min}(D_n D_n^\top)$** To lower bound the minimum eigenvalue of $D_n D_n^\top$, we apply the same argument to the maximum eigenvalue of $(D_n^\top)^{-1} D_n^{-1}$. The largest eigenvalue of $(D_n^\top)^{-1} D_n^{-1}$ is at most

$$n^{-\beta} \log^{-\alpha}(n) \left\| \begin{bmatrix} I_k & 0 \\ K & I_d \end{bmatrix}^{-1} \right\|^2 \left\| \begin{bmatrix} C_n^{1/2} & 0 \\ 0 & \sqrt{\tau^2/\beta} I_d \end{bmatrix}^{-1} \right\|^2$$

Then, $\left\| \begin{bmatrix} C_n^{1/2} & 0 \\ 0 & \sqrt{\tau^2/\beta} I_d \end{bmatrix}^{-1} \right\| \le \max(\|C_n^{-1}\|^{1/2}, \sqrt{\beta/\tau^2}) = \max(\lambda_{\min}(C_n)^{-1/2}, \sqrt{\beta/\tau^2})$.

Again using equation (9) and the triangle inequality, this is at most $O(1)$, meaning the minimum eigenvalue of $D_n D_n^\top$ is at least $\Omega(n^\beta \log^\alpha(n))$.

$\square$

**Proof of Corollary 1:** Suppose by contradiction that there is some $k$ where

$$\sup_{P \in \mathcal{P}_n} P(\|\pi(\beta|H_n) - \mathcal{N}(\hat{\beta}_n, \sigma^2(\mathbf{X}_n^\top \mathbf{X}_n)^{-1})\|_{TV} > c) > k$$

for infinitely many $n$. Then, for all such $n$, there is some sampling rule $\Lambda^n$ whose corresponding distribution trajectory $P$ satisfies

$$P(\|\pi(\beta|H_n) - \mathcal{N}(\hat{\beta}_n, \sigma^2(\mathbf{X}_n^\top \mathbf{X}_n)^{-1})\|_{TV} > c) \geq k.$$

However, this contradicts the statement of Theorem 2.

## E   Heteroskedastic Bandits

---

**Algorithm 2** Heteroskedastic Gaussian bandits

---

**Input** Action set $A = \{1, \ldots, p\}$, Arm sampling rule $\Lambda : (A \times \mathbb{R})^* \to \Delta(A)$, Reward distributions $\mathcal{B}_i = \mathcal{N}(\beta_i, \sigma_i^2)$.
**Output** Sampling trajectory $H_n$

$H_0 \leftarrow \emptyset$
**for** $j = 1, \ldots, n$ **do**
  Sample $a_j \sim \Lambda(\cdot|H_{j-1})$
  Sample $y_j \sim \mathcal{N}(\beta_{a_j}, \sigma_{a_j}^2)$
  $H_j \leftarrow \{(x_1, y_1), \ldots, (x_j, y_j)\}$
**end for**

---

**Theorem 5.** *Suppose we generate a length-$m_n$ trajectory $H_{m_n}^n = ((a_1^n, y_1^n), \ldots, (a_n^n, y_n^n))$ for each $n$ according to decision rules $\Lambda^n$ in the case of Heteroskedastic Gaussian bandits. Assume the sampling procedure satisfies the following conditions:*

*(i)* $\min_i N_{i,n} \xrightarrow{p} \infty$ *where* $N_{i,n} = \sum_{j=1}^n I[a_j^n = i]$.

*(ii)* $\pi(\cdot)$ *is continuous and bounded on $\mathbb{R}^p$ with positive density at $\beta_0$.*

*Then, the posterior distribution $\pi(\beta|H_{m_n}^n)$ satisfies*

$$\|\pi(\beta|H_{m_n}^n) - \mathcal{N}(\hat{\beta}_n, \text{diag}(\sigma_i^2 N_{i,n}^{-1}))\|_{\text{TV}} \xrightarrow{p} 0.$$

*Proof.* We rescale this problem to an instance of the homoskedastic bandit problem and apply Theorem 2. Let the covariate sampling rule be $\tilde{\Lambda}^n(H_{j-1}) = e_{\Lambda^n(H_{j-1})}$ and the rescaled parameters be $\tilde{\beta}_i = \frac{\beta_i}{\sigma_i}$ with the homoskedastic variance $\tilde{\sigma} = 1$. Let $\tilde{\pi}(\tilde{\beta}) \propto \pi(\sigma * \tilde{\beta})$ where $*$ represents element-wise multiplication where $\sigma$ is the vector of $(\sigma_i)_{i=1}^p$. Then, by Theorem 2,

$$\|\tilde{\pi}(\tilde{\beta}|\tilde{H}_{m_n}^n) - \mathcal{N}(\hat{\tilde{\beta}}_n, (\tilde{\mathbf{X}}_n^\top \tilde{\mathbf{X}}_n)^{-1})\|_{\text{TV}} \xrightarrow{p} 0.$$

Rescaling the posterior back to the original parameters gives the desired result.  □

## F   Parametric BvM

There are considerable challenges when extending the BvM result to parametric models satisfying only weak regularity conditions such as differentiability in quadratic mean. Consider replacing the normal distribution in Algorithm 1 with a general parametric model $P_\theta(\cdot|x_j)$ with $\theta = \beta^\top x_j$. We may want to show that in this setting, the posterior distribution is asymptotically normal similar to the BvM statement, i.e.

$$\|\pi(\theta|H_n) - \mathcal{N}(\hat{\theta}_n, J_n^{-1})\|_{\mathrm{TV}} \xrightarrow{p} 0$$

where $J_n = -\sum_{j=1}^{n} \ddot{\ell}_{\hat{\theta}_n}(y_j|x_j)$ is the empirical Fisher Information and $\ell_\theta = \log P_\theta$ is the log-likelihood. Suppose we follow the steps of the classical BvM proof. Letting $\pi^{C_n}(\theta|H_n)$ and $\mathcal{N}^{C_n}(\hat{\theta}_n, J_n^{-1})$ denote truncations of these distributions to the set $C_n$, the classical proof proceeds in the following three steps—we show each of the following convergences:

$$\|\pi(\theta|Y) - \pi^{C_n}(\theta|Y)\|_{\mathrm{TV}} \xrightarrow{p} 0$$
$$\|\pi^{C_n}(\theta|Y) - \mathcal{N}^{C_n}(\hat{\theta}_n, J_n^{-1})\|_{\mathrm{TV}} \xrightarrow{p} 0$$
$$\|\mathcal{N}^{C_n}(\hat{\theta}_n, J_n^{-1}) - \mathcal{N}(\hat{\theta}_n, J_n^{-1})\|_{\mathrm{TV}} \xrightarrow{p} 0.$$

The first step requires a method of truncating the posterior distribution, which is in general quite difficult without strong assumptions on the behavior of the adaptive system similar to the results of [18] and [11]. With independent data, we could truncate the posterior due to Hoeffding bounds on the likelihood, but these bounds do not apply in adaptively collected data. The second step was implied by a second-order Taylor expansion of the log-likelihood in the classical proof, but in adaptive settings, the Fisher information matrix $J_n$ may grow at different rates in different directions, i.e. $\lambda_{\min}(J_n)$ may grow at a different rate than $\lambda_{\max}(J_n)$. This would make an argument based on a Taylor expansion more complicated and a valid proof seems to require further assumptions on the growth rate of $J_n$. Finally, the third step was implied in the i.i.d. setting by the local consistency of $\hat{\theta}_n$, i.e. $J_n^{1/2}(\hat{\theta}_n - \theta_0) = O_p(1)$ but this condition may be violated in adaptive experiments. In fact, as Lai and Wei showed, we require some nontrivial assumptions on the empirical Fisher Information to guarentee consistency, even in the case of Gaussian regression.

## G   Real-world example

We compute the BvM total variation distance on a real-world instance of Bernoulli Thompson sampling provided in [28]. The dataset consists of the interaction of a fashion recommendation algorithm with users, where the algorithm is optimizing for the number of clicks on fashion items. Although the original dataset is modeled with a contextual bandit setting with a batch size of 3, we simplify their environment to an 80-armed bandit problem by disregarding the context and flattening the batch size. These simplifications were made solely for computational convenience. The dataset consists of 12m impressions or time steps, during which we update the posterior distribution with a Beta$(1, 1)$ prior. The BvM total variation distance throughout this rollout is shown below.

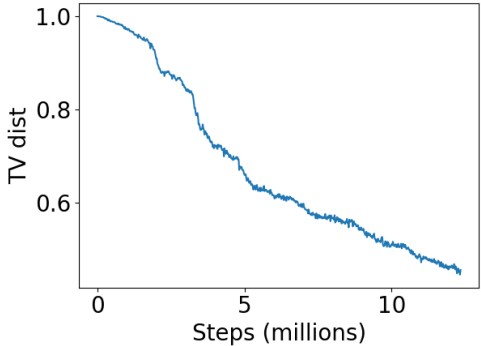

Figure 4: BvM TV distance for the Zozo dataset with prior Beta$(1, 1)$. TV estimates computed with $10^4$ samples and have standard error at most $0.004$.

As seen in Figure 4, the convergence of the total variation distance is quite slow. This is perhaps due to the sparse success rates in the Bernoulli bandit and the larger dimensionality of the parameter

space. It should be noted that the original dataset contained missing data within batches, i.e. not all batches had size 3.

## H  Plots

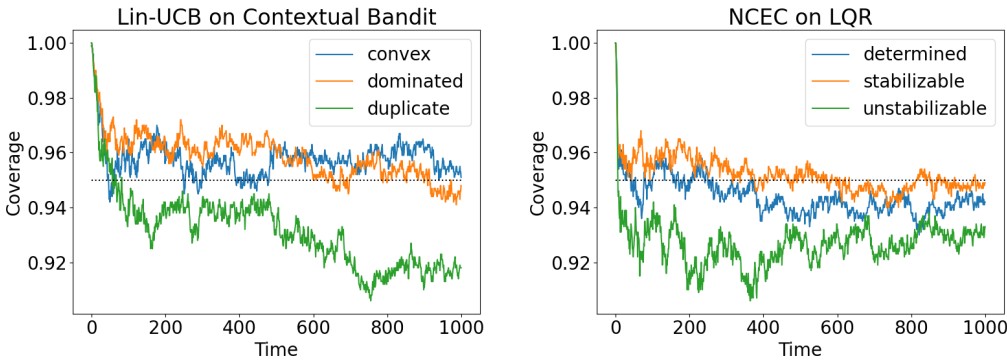

Figure 5: Frequentist coverage of the Bayesian credible interval for the contextual bandit and LQR, under the same configurations as Section 6. Coverage estimates shown have standard error at most 0.004.

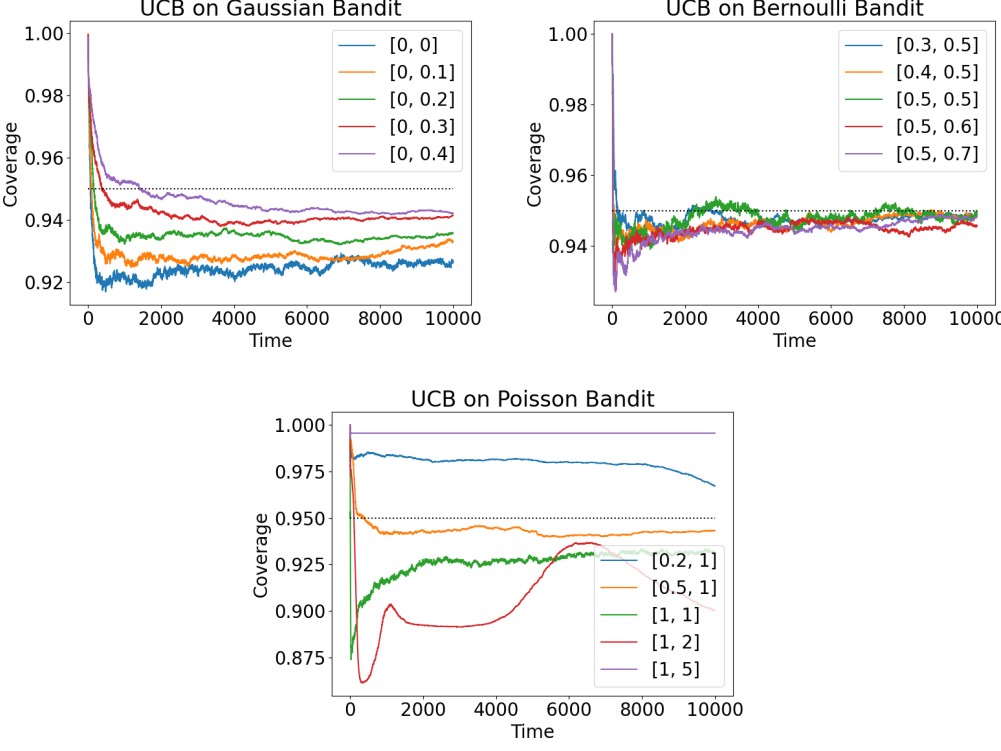

Figure 6: Frequentist coverage of the Bayesian credible interval for UCB on Gaussian bandits, Bernoulli bandits and Poisson bandits, under the same configurations as Section 6. Coverage estimates shown have standard error at most 0.004.

