# OpenReview forum: "Bernstein–von Mises for Adaptively Collected Data"
_NeurIPS.cc/2025/Conference — NeurIPS 2025 poster_

### Official Review · Reviewer_7V5X · 2025-07-02

**Clarity:** 3
**Significance:** 2
**Originality:** 3
**Rating:** 5
**Confidence:** 4

**Summary:**

This paper presents Bernstein-von Mises (BvM) results for settings in which data collection is adaptive. Traditional BvM results will often operate under assumptions that are not entirely aligned with adaptive contexts, like assuming iid data, so the results presented here are a kind of extension, with some particular emphasis on applicability to specific types of Linear Gaussian and Exponential Family bandit settings. Further, they claim to have achieved these results without assuming traditional stability conditions that would otherwise ensure "frequentist-validity", and yet "bayesian-validity" is still achieved. Finally, numerical experiments examine the nature of the convergence that is guaranteed by these results.

**Questions:**

# Question 1:
The authors write *"$\ldots$ Lai and Wei showed that condition (i) of Theorem 1 is necessary for MLE consistency (which our proof relies on), as there are counterexamples where the condition is only marginally violated but consistency fails [20, Example 1]."*

This line is rather confusing. If I'm understanding correctly the literal meaning of this statement, the authors are indicating that MLE consistency implies condition (i) because of the existence of counterexamples? There is something very imprecise here, considering later on, the authors also dive into an outlined hypothetical: *"Second, even if $\hat{\beta}_n$ is consistent but condition (i) fails, it may not be the case that $\ldots$", which their earlier statement, if taken literally, would preclude the possibility of.

As far as I understand, Lai and Wei actually show that condition (i) (and other assumptions) suffice for there to be (strong) consistency, and then their cited example is meant to show that condition (i), in their words,*"is in some sense weakest possible."* So I'm guessing the authors actually meant to say something like, Lai and Wei showed that condition (i) of Theorem 1 is *nearly* necessary for MLE consistency.

# Question 2:
The restriction to a Gaussian statistical model, as outlined in Section 3 $\ldots$ can the authors speak on the extent that their analysis relies on this?

**Ethical Concerns:**

["NO or VERY MINOR ethics concerns only"]

**Final Justification:**

My initial assessment was already positive (see above), though I pointed out that the appendix section containing all the proofs did not appear attached. The authors kindly pointed me to the separately zipped supplementary folder containing the appendix. I have since reviewed the proofs in the appendix.

If I'm not mistaken there are some minor typos (e.g. line 424), but they didn't get in the way of understanding the arguments. And I now raise the overall score from 4 (Borderline accept) to 5 (Accept).

**Limitations:**

yes

**Paper Formatting Concerns:**

Line 92: and hence the task at hand **is to** perform

**Quality:**

3

**Strengths And Weaknesses:**

I'm not entirely sure what to make of the fact that this theory paper has a missing appendix and hence no proofs. This leaves me on the fence, and I am preliminarily submitting a borderline rating that leans towards accept (I'm trying to be optimistic), and will look forward to the rebuttal phase and the comments/guidance from the chair.

# Strengths:
- There are some helpful, high-level explanations in the main body of the role that various hypotheses and conditions ( e.g. condition (i) ) have in the theorems. There are also informative summaries of how the proofs (though missing) proceed. As well, I appreciated the examples, remarks and discussions surrounding the possibility of finding frequentist-invalidity but Bayesian-validity.
- The topic is motivated, and incorporates some Bandit settings

# Weaknesses:
- Appendix appears to be missing $\ldots$ so no proofs provided in this manuscript. However, this omission of the Appendix seems to be accidental. A bit awkward to assess the level of theoretical contribution without the proofs. There are certainly numerical experiments that illustrate the Bayesian validity, but this work is a theory paper.
- The authors largely restrict to a Linear Gaussian observation scheme

---

> ### Author Rebuttal · Authors · 2025-07-30
>
> We apologize for any confusion regarding the appendix section. We included it in the supplementary material - after unzipping the file, there should be a folder containing a file named “appendix.pdf” and a folder named “code”. The appendix may be found in the “appendix.pdf” file. These files are showing in our system as having been uploaded and when we download them ourselves, they appear as we expect, but let us know if there is an issue with accessing our appendix and we can try to fix it. We believe that the supplementary material was accessible to at least some of the other reviewers, as we have received comments from other reviewers regarding our proofs (which appear in the appendix).
>
> We briefly discuss in Appendix F reasons why our proof technique does not extend to the setting of general parametric models; please let us know if our discussion in Appendix F fails to clarify the reason that our proof technique does not generalize. We apologize if this was not accessible for you when you reviewed our paper.
>
> Thank you for catching this issue in Question 1! We should have written it differently, and we will be sure to clarify in our revision what we meant to say: “First, Lai and Wei showed that condition (i) of Theorem 1 is _nearly_ necessary for MLE consistency. In particular, if condition (i) of Theorem 1 is removed, the statement becomes false.”
>
> Thank you for pointing out the formatting mistake on line 92 - it will be corrected.

---

> > ### Comment · Reviewer_7V5X · 2025-08-04
> > **Raising Score from 4 to 5**
> >
> > My initial assessment was already positive (see above), though I (mistakenly) pointed out that the appendix section containing all the proofs did not appear attached. The authors kindly pointed me to the separately zipped supplementary folder containing the appendix. I have since reviewed the proofs in the appendix.
> >
> > I raise my overall score of the submission from 4 (Borderline accept) to 5 (Accept).

---

### Official Review · Reviewer_DzMC · 2025-07-02

**Clarity:** 3
**Significance:** 3
**Originality:** 3
**Rating:** 4
**Confidence:** 3

**Summary:**

This study investigates the Bernstein–von Mises (BvM) theorem for adaptively collected data. Uncertainty quantification is a central issue in statistical analysis, and the authors aim to use the BvM theorem to bridge Bayesian and frequentist perspectives in this setting.

**Questions:**

See above.

**Ethical Concerns:**

["NO or VERY MINOR ethics concerns only"]

**Final Justification:**

Personally, I am a bit skeptical about the results of the paper. However, I do not have any concrete evidence to support this view. My intuition is that the uncertainty could be assessed using a martingale central limit theorem or a concentration inequality, and when these do not hold, obtaining a BvM-type guarantee—while supported from a Bayesian perspective—may not yield practically useful bounds. That said, this is merely my personal intuition, so please do not put much weight on it. Since there does not appear to be any fatal error in the proofs, I will stick to my original score and vote for a borderline accept.

**Limitations:**

N/A.

**Paper Formatting Concerns:**

None.

**Quality:**

3

**Strengths And Weaknesses:**

I recognize that the authors tackle ambitious tasks in adaptive experimental design. I also agree that uncertainty quantification for adaptively collected data is important, and the manuscript seeks to provide a general and powerful tool for this purpose.

Nevertheless, I encountered several difficulties while reading the draft, listed below.
1. What are the precise definitions of $\Lambda$, $\lambda_{\min}$, and $\lambda_{\max}$?
2. What is the formal definition of the sampling rule?
3. Do the assumptions required for each theorem hold?
(If I have overlooked these definitions, I apologize to the authors.)

Among these points, I believe the third is the most critical. The authors remark that suboptimal arms are pulled only $O_p(\log n)$ times. I agree that this assumption is central to the field, and, in practice, it renders many methods unrealistic. I would like to understand how the authors address this issue.

For example, Hadad et al. (2021) and subsequent work by the same group show that asymptotically normal estimators can be obtained under appropriate assumptions in bandit problems. However, when I examine their theorems, I find that the convergence rate becomes $1/\sqrt{\log(n)}$ for optimal algorithms, implying that, relative to the usual $1/\sqrt{n}$ rate, we would need $\exp(n)$ samples to achieve comparable performance for normal approximation. I consider this requirement unrealistic.

Although I cannot completely follow the notation, I suspect that a similar convergence rate is implicitly required through $\lambda_{\min}$. If so, I ask the authors to clarify and justify the assumptions. If not, I would appreciate an explanation of how they avoid the problem described above.

---

> ### Author Rebuttal · Authors · 2025-07-30
>
> To clarify our notation, we will add the sentence “Let $\lambda_{\text{min}}(A)$ and $\lambda_{\text{max}}(A)$ denote the minimum and maximum eigenvalues of the matrix $A$ respectively” in the introduction of our paper. We will also clarify at the beginning of section 3 that $\Lambda$, our sampling rule, is allowed to be an arbitrary function of the history up to the current data point; i.e. it is meant to denote a placeholder that can represent any sampling algorithm including UCB, Thompson sampling, autoregressive models, etc.
>
> We included Theorem 2 as a separate theorem specifically to address the setting of multi-armed bandits, which does not satisfy the assumptions of our main theorem. Even though the assumptions of Theorem 2 are satisfied in the bandit setting, the convergence provided by our theorem may be logarithmically slow if suboptimal arms are pulled at logarithmic rates. In other words, we don’t avoid the problem you identified, and we will clarify in our paper after our main theorem that the convergence can be very slow.

---

> > ### Comment · Reviewer_DzMC · 2025-08-05
> >
> > Thank you for your response. I am satisfied with the answers provided to my questions.
> >
> > To be honest, I do not view the contribution of this paper as particularly strong. In the context of adaptive experimental design, it is often possible to utilize properties such as martingales, which allow the use of concentration inequalities and asymptotic theory. I remain somewhat skeptical as to whether the situation in which the authors’ approach is truly necessary arises in practice.
> >
> > That said, my reservations about the practical usefulness of the method are inherently subjective, and I do not believe it would be appropriate to base my overall assessment on them. At the very least, the theoretical framework is sound. For these reasons, I would like to maintain my current score, albeit for somewhat reserved reasons.

---

> > > ### Author Response · Authors · 2025-08-06
> > >
> > > Thank you for your feedback. We agree that martingale central limit theorems and concentration inequalities have provided quite general methods to obtain frequentist-valid inference in a variety of settings, but we emphasize that these results are of a fundamentally different nature than ours. Indeed, in our background section we discuss the stability condition as a key factor in using these martingale properties to prove asymptotic normality of the maximum likelihood estimator, and indeed one of the interesting aspects of our results is that they can hold even when these martingale-based results do not.
> > >
> > > We would like to point out that we do not propose a new method to be used in practice when the stability condition fails, but instead study a widely used approach in adaptive experimentation, which is Bayesian UQ. In many non-adaptive settings, one strong justification for using such UQ is that it is also asymptotically frequentist-valid (by standard i.i.d. BvM and asymptotic normality of the MLE), and our paper studies whether this is the case in adaptive data. We find that the answer can be “no” in some reasonable adaptive settings, but, perhaps surprisingly, we prove this by showing that BvM does hold quite generally, while other works have shown that asymptotic normality of the MLE does not [1,2], and hence Bayesian UQ is not asymptotically frequentist valid in general. Put another way, while Bayesian UQ is always Bayesian-valid and we show BvM holds under very weak assumptions on the adaptivity, this does not provide a fix for the well-known issue with frequentist validity for adaptive data when stability fails to hold. Thus our result urges caution for practitioners who use Bayesian UQ in adaptive settings and expect it to be frequentist valid in an asymptotic sense.
> > >
> > > [1] Hadad, Vitor, et al. "Confidence intervals for policy evaluation in adaptive experiments." Proceedings of the national academy of sciences 118.15 (2021): e2014602118.
> > >
> > > [2] Kelly Zhang, Lucas Janson, and Susan Murphy. Inference for batched bandits. Advances in neural information processing systems, 33:9818–9829, 2020.

---

### Official Review · Reviewer_RwF3 · 2025-07-03

**Clarity:** 1
**Significance:** 2
**Originality:** 3
**Rating:** 5
**Confidence:** 3

**Summary:**

This paper extends the Bernstein-von Mises (BvM) theorem to the adaptive data collection setting, in which the data is non-i.i.d. and the ratio between maximum and minimum eigenvalues of the Fisher information matrix is unbounded. It establishes that under such settings, with mild eigenvalue growth conditions, BvM-analogous convergences hold, implying Bayesian credible intervals retain frequentist validity asymptotically, even under adaptive data collection. The authors prove BvM results for adaptive linear Gaussian models, multi-armed bandits, and contextual bandits, supported by simulations validating convergence and coverage properties.

**Questions:**

Bayesian UQ can match frequentist UQ asymptotically, yet be frequentist-invalid when stability fails

1. The bounded $b^{\prime\prime\prime}$ requirement excludes some exponential families (e.g., Poisson near zero). Is this necessary? Can the proof be adapted without boundedness?
2. Can results be validated on a public adaptive dataset? If not, please discuss challenges.
3. Theorem 1 claims Bayesian validity under prior misspecification, but how quickly does this "washout" occur? Simulations demonstrate slow TV convergence for large margins. Please discuss the implications for finite-sample safety.
4. The relaxation of the usual stability condition doesn't seem to be as significant as presented, since we still need it to guarantee frequentist-validity, or am I missing something?

**Ethical Concerns:**

["NO or VERY MINOR ethics concerns only"]

**Final Justification:**

The authors duly addressed my concerns during the rebuttal and enhanced the clarity. The paper is solid with potential impact on future research. I raise my score from 3 to 5.

**Limitations:**

Suggestions:
1. Discuss mitigations for instability-induced invalidity.
2. Address finite-sample prior sensitivity observed in simulations.
3. Expand societal impact: Could invalid UQ in adaptive trials lead to harmful decisions?

**Quality:**

3

**Strengths And Weaknesses:**

**Strengths**
- First BvM extension to adaptive data, addressing a critical gap in uncertainty quantification for bandits/RL.
- Broad applicability to Gaussian bandits, exponential families, and contextual settings.
- Theorems are rigorously proven with careful handling of non-i.i.d. likelihoods and anisotropic Fisher information.
- Simulations validate positive/negative results across diverse adaptive settings (MAB, contextual bandits, LQR).

**Weaknesses**
1. Clarity issues:
- The umbrella term *uncertainty quantification (UQ)* is used throughout. The author should use explicit terms of *Bayesian credible intervals* and *frequentist confidence intervals*.
- Jargons like "Wald-type", "frequentist-valid" are used without first establishing precise definitions.
- Using $H_n^n$ for triangular arrays (Section 4) alongside $H_n$ for fixed-$n$ trajectories is confusing
2. Technical concerns:
- Proof sketches (Section 3) omit key steps (e.g., why truncation works without bounded condition numbers)
- Convergence in TV implies pointwise agreement but may not ensure uniform frequentist validity. Authors should discuss implications for confidence sets.
3. Experimental limitations:
- All experiments use synthetic data. No validation on real adaptive datasets.
- Coverage plots are only shown for MAB. For contextual bandits/LQR, only TV distance is reported, leaving coverage unexplored.

---

> ### Author Rebuttal · Authors · 2025-07-30
>
> **Clarity**:
>
> We chose to use the term “uncertainty quantification” in the paper because we wanted to include a variety of different use cases, not just constructing confidence/credible intervals (although these are what our empirical evaluations focus on). For instance, frequentist asymptotic normality is often used for forms of UQ other than confidence interval construction such as (multiple) hypothesis testing or prediction intervals (i.e., a high probability region for the response given a new test X). In the Bayesian paradigm, posterior distributions (and approximations thereof) are also often used for hypothesis testing and constructing posterior predictive intervals. To clarify our terminology, we will add the sentence “We will use the term frequentist (resp. Bayesian) UQ generically to refer to any of the many forms of frequentist (resp. Bayesian) statistical inference such as hypothesis tests, confidence intervals, or prediction intervals (resp. such as Bayesian hypothesis tests, credible intervals, or posterior predictive intervals).” to the introduction of the paper. We would prefer not to directly replace the phrase “UQ” as suggested, as that would restrict the use cases to frequentist confidence intervals and Bayesian credible intervals.
>
> We will add precise definitions of “Wald-type” and “frequentist-valid” in the introduction section of our paper after the “Contributions” paragraph: “By Wald-type frequentist UQ, we refer to standard asymptotic hypothesis testing or confidence interval construction using the MLE and its asymptotic Normality—for instance, as demonstrated in Example 15.6 of [1].” and “Throughout this paper, we refer to a confidence set as asymptotically frequentist-valid at level alpha if the limit inferior of the frequentist coverage is at least 1-alpha for any parameter value. Similarly, we refer to a credible set as asymptotically Bayesian-valid at level alpha if the limit inferior of the Bayesian coverage is at least 1-alpha.”
>
> Regarding the notations $H_n^n$ and $H_n$, could the reviewer please elaborate on what specifically they found confusing? We would be happy to change our notation accordingly, but weren’t sure what aspect the reviewer was referring to. In case it helps, the reason we chose these notations was as follows: we use $H_j$ (e.g., in Algorithm 1) to denote the history (hence the letter $H$) up to time $j$. In the non-triangular array setting this history extends as the subscript increments, i.e., $H_{j+1}$ is simply $H_j$ with one more time step concatenated, so there is only one sequence of histories. But in the triangular array setting, the sequence of histories can be completely different for different $n$, i.e., $H_j^n$ and $H_j^{n+1}$ are not the same, so we need notation to distinguish them, hence the superscript so that $H_j^n$ and $H_j^{n+1}$ have distinct notations. Then, when we look at the entire history up to time $n$, which is an important quantity, the subscript also takes the value $j=n$, and we get the notation $H_n^n$, meaning the history up to time step $n$ of the $n$th trajectory in the triangular array.
>
> **Technical Concerns**:
>
> We will add a more detailed explanation of why truncation works without bounded condition numbers to the extent that space allows in our main paper, akin to: “Our proof for why we can truncate the posterior relies critically on the second part of condition (i) of Theorem 1 which allows us to show that the density of the representative normal distribution outside of local ellipsoids converges to zero. We furthermore show that the posterior can be written as proportional to the prior multiplied by the density of the representative normal, meaning we can truncate the distribution to a local ellipsoid even if the prior is unbounded.”
>
> It is correct that our results are not uniform in the underlying parameter. However, we have not come across any uniform BvM results in the existing literature (even in the standard i.i.d. setting). Additionally, it is generally not the case that the asymptotic normality of the MLE is uniform in the underlying parameter. For example, the (square-root-Fisher-information-rescaled) MLE is not asymptotically Normal in a triangular array setup with p=1/n for IID Bernoulli data. Consequently, the utility of a parameter-uniform BvM statement is unclear as, in most applications, it would not imply uniform frequentist validity of Bayes’ procedures.
>
> Your point about uniformity does raise an interesting question, however, which we had not considered before, which is whether or not our result gives uniform convergence of the TV distance over suitable sets of sampling rules. More formally, the triangular array result Theorem 2 of our paper can be interpreted as the following non-triangular-array uniform convergence result for multi-armed bandits.
>
> Corollary 1. For any sequences $(r_n, \epsilon_n)$ for which $r_n \to \infty$ and $\epsilon_n \to 0$, let $\mathcal{P}\_n$ be the sequence of sets of distributions $P$ on $H_n$ induced by sampling rules $\Lambda$ satisfying $P(\lambda_{\text{min}}(X_n^\top X_n) > r_n) > 1-\epsilon_n$. Then, we have for any $c>0$,
>
> $$\limsup_{n \to \infty} \sup_{P \in \mathcal{P}\_n} P(||\pi(\beta|H_n) − \mathcal{N}(\hat{\beta}\_n, \sigma^2(X_n^\top X_n)^{−1})||_{TV} > c) = 0$$
>
> This shows that we do have uniform convergence of TV distance over sets of sampling rules where $\lambda_{\text{min}}(X_n^\top X_n)$ grows at some uniform rate. We will add this to our revision as a remark after Theorem 2.
>
> **Experimental Limitations**:
>
> Our validation procedure consists of running multiple iterations of an experiment on synthetic data and comparing confidence intervals to the ground truth to estimate coverage and other quantities of interest. This process would be difficult to execute in real world applications where we often don’t have access to the ground truth (perhaps only estimates of it). We can, however, include examples of real-world data and report the TV distance between the Bayesian posterior and the MLE-centered Gaussian distribution (with variance-covariance matrix using the empirical Fisher information matrix, rather than the true one, as the latter is unknown to us). However, it should be noted that, as we established, Bayesian credible intervals may fail to be asymptotically frequentist-valid (in the same settings in which Wald-type intervals are asymptotically valid).
>
> Showing coverage plots for the other non-MAB environments is a great suggestion - we’ll be sure to add these plots to the Appendix (due to space limitations) in the revision.
>
> **Questions**:
>
> We apologize that we did not specify this more clearly earlier, but the bounded $b’’’$ condition is actually satisfied for all natural exponential family models (provided that the $\beta_{0,i}$ parameters are in the interior of the parameter space). This is because the log-partition function $b(\eta)$ is smooth on the interior of the natural parameter space (see, e.g., Theorem 2.2 of [2]) and consequently it is bounded on any sufficiently small neighborhood of any interior point.
>
> In the case of Poisson, for example, letting $\eta=\log \lambda$ be the natural parameter of the Poisson distribution, we have $b(\eta)=\exp(\eta)$ meaning $b’’’(\eta) = \exp(\eta)$, which is bounded on a union of neighborhoods around a finite number of true parameters. We will replace condition (i) with just “$\beta_{0,1},\ldots,\beta_{0,p}$ are in the interior of the natural parameter space”.
>
> The “washout” effect can indeed be logarithmically slow if the minimum eigenvalue of the Fisher Information grows logarithmically, which is certainly a problem for finite-sample validity which we agree should be better acknowledged. We will add some text discussing this to the last paragraph of Section 3: “Note, however, that the rate at which the ‘wash out’ effect occurs depends on the rate at which the posterior distribution converges in TV distance to a normal distribution, which may be logarithmically slow for optimal bandit algorithms. Thus, it may not be accurate in finite samples to treat a misspecified prior as having `washed out’.”
>
> Yes, if we want to show asymptotic frequentist-validity using our theory, we do need the MLE to be asymptotically normal, which can be guaranteed by regularity conditions such as stability. What’s surprising is that the step that allows us to conclude asymptotic agreement between Bayesian inference and Wald-type frequentist inference, i.e. the BvM statement, still works without the stability condition. In this sense, our result is not a relaxation of the stability condition on frequentist results but instead a new theorem about the behavior of Bayesian inference from a frequentist perspective.
>
> **Suggestions**:
>
> Our work does not provide a mitigation to instability-induced instability, and as far as we know, there is no Bayesian mitigation for this issue. We will revise our Societal Impact section by addressing the fact that Bayesian UQ may be used as if it were frequentist valid. We will include a warning that in general, Bayesian UQ should not be treated as frequentist valid in adaptive settings.
>
> [1] Aad W Van der Vaart. Asymptotic statistics, volume 3. Cambridge university press, 2000.
> [2] Brown, Lawrence D. "Fundamentals of statistical exponential families: with applications in statistical decision theory." Ims, 1986.

---

> ### Comment · Reviewer_RwF3 · 2025-08-05
>
> I appreciate the author's detailed response. I believe that most of my concerns have been duly addressed. However, there are still a few points as follows:
> - Regarding the notation $H_n^n$, my apologies for not being clearer. I meant to suggest using a different index for the length of the sequence, as it doesn't seem the sequence length needs to coincide with the maximum timestep $n$ (or am I wrong about this?)
> - Regarding Theorem 3's statement, I think it is appropriate to make condition (i) more succinct and further elaborate on the implication in the proof. However, I suggest the authors also acknowledge in the paper that the result is limited to "the interior of the natural parameter space" and hence will not hold in some intuitive extremes, such as $Pois(\lambda \to \infty)$ (near continuous, which I mistook with near zero in the initial review)
> - Regarding the delivery of the main contribution, I suggest that the authors thoroughly revise several details to distinguish "asymptotic agreement" from "frequentist validity". For instances, some of the current sentences could lead to confusion:
>     - Lines 14-16: "Our results **do not require the standard stability condition for validity of Wald-type frequentist UQ**, and thus provide positive results on frequentist validity of Bayesian UQ under stability".
>     - Lines 59-60: "...surprising aspect of our BvM results is that they **do not require the key stability condition required for asymptotic validity** of Wald-style frequentist inference..."

---

> > ### Author Response · Authors · 2025-08-06
> >
> > * Thank you for clarifying your comment about notation–we agree that the sequence length does not need to correspond to the index of the trajectory. We will change the sequence length to $m_n$ and the notation of the trajectory to $H_{m_n}^n$. We will modify the proof of Lemma 1 and Theorem 2 to make it consistent with this new theorem statement.
> >
> > * In addition to changing the wording of Theorem 3’s statement, we will be sure to add a sentence to acknowledge that the result is limited to the interior of the natural parameter space and hence does not apply to extremes like Poisson near infinity or Bernoulli near 1.
> >
> > * Thank you for pointing out that our original wording in these phrases was confusing. We will change these sentences to:
> > 	* Lines 14-16: “Our result showing this asymptotic agreement does not require the standard stability condition required by works studying validity of Wald-type frequentist UQ; in cases where stability is satisfied, our results combined with these prior studies of frequentist UQ imply frequentist validity of Bayesian UQ.”
> > 	* Lines 59-60: “A surprising aspect of our BvM results is that they do not require the key stability condition used in frequentist validity results; however, it is known that Wald-type frequentist inference may fail to be asymptotically valid in cases when the stability condition does not hold.”

---

> > > ### Comment · Reviewer_RwF3 · 2025-08-08
> > >
> > > I am happy to raise my score from 3 to 5, given the enhanced clarity. The paper is solid with potential impact on future research.

---

### Official Review · Reviewer_VNGk · 2025-07-03

**Clarity:** 3
**Significance:** 2
**Originality:** 2
**Rating:** 5
**Confidence:** 4

**Summary:**

The authors derive a BvM theorem in a parametric setting for data that is sequentially observed in a linear model. Specifically, a data stream $H_i = ((y_1,x_1),\dots,(y_i,x_i))$ where $x_{i+1}$ can be chosen on the basis of $H_i$, after which $Y_{i+1} = \beta^\top x_{i+1} + noise$ is observed, with iid Gaussian noise with known variance.

The authors show a BvM theorem for $\beta$, comparing the posterior to a normal distribution centered at the MLE and with (empirical) Fisher information as its variance.

**Questions:**

Is there a reason why the proof technique would not work in the typical generality of parametric BvM's? I.e. likelihoods that differentiable in quadratic mean?

**Ethical Concerns:**

["NO or VERY MINOR ethics concerns only"]

**Limitations:**

The main limitation in my eyes is mentioned in the strength/weaknesses section of the review.

**Quality:**

3

**Strengths And Weaknesses:**

Strengths:

* The paper is well written.
* The topic is interesting.
* I like how the link to bandits is fleshed out in the paper.
* The proofs appear sound.

Weaknesses:
* The setting is very limited; I see no reason why the proof technique would not work in the typical generality of parametric BvM's and there is a lack of discussion concerning this in the paper.
* There is a lack of literature overview / comparison of BvM's with non-i.i.d. data. How does the result relate to investigations such as [1]? Does general (non-iid) machinery as developed by [2] not apply?

[1] J. D. Borwanker, G. Kallianpur & B. L. S. Prakasa Rao. The Bernstein–von Mises Theorem for Markov Processes. Annals of Mathematical Statistics
[2] Ismaël Castillo & Judith Rousseau. A Bernstein–von Mises theorem for smooth functionals in semiparametric models. Annals of Statistics
[2]

---

> ### Author Rebuttal · Authors · 2025-07-30
>
> We have spent considerable effort thinking of ways to extend our results to more general parametric models. We discuss in Appendix F reasons why our proof technique does not extend to the setting of general parametric models. Briefly, the main reason is that in general parametric models, we require an accurate second-order Taylor expansion of the log likelihood, but the unbounded condition numbers of the Fisher information matrix make it difficult to give a sufficiently strong error bound (even with the assumption of differentiability in quadratic mean). Please let us know if our discussion in Appendix F fails to clarify the reason that our proof technique does not generalize. Also, we’ll be sure to explicitly state the goal of generalizing to parametric models that are differentiable in quadratic mean, as we currently do not mention q.m.d. in Appendix F (though this class of models was exactly what we had in mind in our attempts to generalize).
>
> We compared our results with previous BvM results in adaptively collected data in the second paragraph of our “Background” section. Prior results assume that the ratio of the maximum and minimum eigenvalues of the Fisher information matrix are bounded, and in fact most assume the stronger condition that there exists some consistent norming rate $\delta_n \to 0$ where $\delta_n I_n$ converges in probability to some limiting Fisher information $I$. The results established by [2] also make this assumption where this norming rate is $1/n$, as the posterior distribution is shown to be asymptotically normal with variance $V_0/n$ for some fixed $V_0$. The assumption of a bounded condition number is not satisfied by optimal bandit algorithms as discussed in section 2 of our paper, and therefore both our theorem statement and proof technique do not require this assumption. To our knowledge, our result is the first non-i.i.d. BvM result to accomplish this. For the sake of completeness, we will add [2] to our block of citations on line 82. The result given in [1] provides an asymptotic error bound on the BvM distance, providing a more quantitative analysis of the convergence rate, similar to other prior works such as [3] which gives a nonasymptotic error bound. Their argument, like ours, requires truncating the posterior distribution, which they argue by proving bounds on the observed likelihood and empirical Fisher information. However, their argument relies critically on Doeblin’s condition which guarantees a fixed rate of convergence of the Markov chain to a stationary distribution, so it is difficult to generalize their result to non-Markovian settings such as bandits.
>
> [3] Lee, Jeyong, Junhyeok Choi, and Minwoo Chae. "Online Bernstein-von Mises theorem." arXiv preprint arXiv:2504.05661 (2025).

---

### Official Review · Reviewer_iYuq · 2025-07-13

**Clarity:** 3
**Significance:** 2
**Originality:** 3
**Rating:** 4
**Confidence:** 3

**Summary:**

This paper addresses the challenge of uncertainty quantification (UQ) for adaptively collected data, which arises in settings such as adaptive experiments, multi-armed bandits (MABs), and reinforcement learning. The core contribution is an extension of the classical Bernstein-von Mises (BvM) theorem to adaptively collected data, establishing asymptotic equivalence between Bayesian UQ and Wald-type frequentist UQ in these complex scenarios. One key takeaway is that Bayesian UQ can be asymptotically valid in a Bayesian sense even in adaptive settings, but asymptotic frequentist validity is now always assured — especially when the stability condition fails. In other words, posterior credible intervals can be misleading from a frequentist perspective, even though they are valid in a Bayesian sense (Bayesian validity).

**Questions:**

No further questions.

**Ethical Concerns:**

["NO or VERY MINOR ethics concerns only"]

**Limitations:**

Yes.

**Paper Formatting Concerns:**

No formatting concerns.

**Quality:**

3

**Strengths And Weaknesses:**

1. Strengths:

(S1) The proposed BvM theorem for adaptively collected data rigorously establishes when Bayesian UQ aligns with Wald-type frequentist UQ in adaptive settings, addressing a major gap in theoretical statistics.

(S2) The main results are proven under mild conditions, notably not requiring the stability condition often assumed in prior works (e.g., [1]), thereby broadening the range of applicable scenarios.

(S3) The paper provides a clear distinction between Bayesian validity vs. frequentist validity: while Bayesian UQ is always asymptotically valid in a Bayesian sense as long as the prior “washes out”, the asymptotic frequentist validity is not always guaranteed — especially when the stability condition fails. This distinction is crucial for practitioners.


2. Weaknesses:

(W1) Even though the BvM theorem is shown to hold without the stability condition, statistical inference with frequentist validity is not ensured in these cases. The paper demonstrates that posterior credible intervals can fail to have correct frequentist coverage without the stability condition.

(W2) The author didn’t discuss the exact meaning of some key terminologies including asymptotic frequentist validity and Bayesian validity in the paper. While statisticians may be familiar with these terms, many researchers in ML or artificial intelligence might not be, so I believe it would be helpful to provide a direct explanation.


[1] Tze Leung Lai and Ching Zong Wei, “Least squares estimates in stochastic regression models with applications to identification and control of dynamic systems”, The Annals of Statistics, pp. 154-166, 1982.

---

> ### Author Rebuttal · Authors · 2025-07-30
>
> (W1) It is correct that our results imply that, in some cases, Bayesian credible intervals will fail to be asymptotically frequentist-valid when frequentist MLE-based inference is invalid (e.g., possibly because the stability condition is not satisfied). However, instead of a weakness, we view this as an interesting consequence of our work. In particular, our main contribution is not a new method of implementing frequentist-valid inference, but rather to explain the behavior of Bayesian inference from a frequentist perspective. We note in our “Discussion and Future Work” section that asymptotic frequentist validity may fail even when the BvM statement holds, and we discuss potential ways a practitioner could mitigate this issue. For the sake of clarity, we will add the sentence “Note that this paper does not suggest a method of asymptotically frequentist-valid Bayesian inference; in fact, we would like to warn practitioners against using either Wald-type frequentist UQ or Bayesian UQ as if it were asymptotically frequentist-valid.” to the “Discussion and Future Work” section.
>
> (W2) We will give explicit definitions for asymptotic frequentist validity and asymptotic Bayesian validity in the introduction by adding the following sentences to a “Notation” paragraph after the “Contributions” paragraph in section 1: “Throughout this paper, we refer to a confidence set as asymptotically frequentist-valid at level alpha if the limit inferior of the frequentist coverage is at least 1-alpha for any parameter value. Similarly, we refer to a credible set as asymptotically Bayesian-valid at level alpha if the limit inferior of the Bayesian coverage is at least 1-alpha.”

---

### Note · Authors · 2025-08-12

We would like to thank our five reviewers for their detailed comments and suggestions on our work. Here, we would like to briefly summarize the main strengths of our work identified by our reviewers as well as how we will address our reviewers’ concerns and suggestions.

The main theorem of our work extends the Bernstein von Mises theorem to adaptively collected data while requiring minimal conditions, allowing for general applications in new challenging settings such as exponential-family bandits, contextual bandits, and linear quadratic regulators. Our reviewers note that our work stresses the difference between frequentist and Bayesian inference by providing negative examples where Bayesian credible intervals fail to be asymptotically frequentist-valid. Broadly speaking, our reviewers attest to the rigor of our proofs and the usefulness of intuitive explanations surrounding our main theorems in the text.

Our reviewers kindly pointed us to several clarity issues in our paper and we have given detailed plans in our responses to define our terminology and fix unclear or ambiguous wording. Our reviewers also gave suggestions for expanding on our work, such as by including additional plots, adding new references, adding to our proof sketches, and discussing a corollary of one of our theorems. We will include all of these in the main text to the extent to which space allows, and otherwise include the remaining additional material in the appendix.

---

### Decision · Program_Chairs · 2025-09-17

**Decision:**

Accept (poster)

**Comment:**

The paper proves a Bernstein-von Mises theorem for the regression coefficients in a parametric linear-Gaussian model where covariates are chosen according to an arbitrary function. It then applies these results to Bayesian uncertainty quantification in Gaussian multi-armed bandits and adaptive Gaussian linear bandits.

Reviewers found the paper well-written, with mathematically sound proofs, useful bandit applications, and reasonably mild assumptions. They noted limitations stemming from the reliance on linear-Gaussian structure and a relatively limited discussion of BvM in non-i.i.d. settings. The authors’ rebuttal addressed these concerns convincingly. I recommend acceptance.